# Data assimilation of radar reflectivity volumes in a LETKF scheme

Thomas Gastaldo[1,2], Virginia Poli[1], Chiara Marsigli[1], Pier Paolo Alberoni[1], and Tiziana Paccagnella[1]

[1]Arpae Emilia-Romagna Hydro-Meteo-Climate Service - Bologna, Italy
[2]University of Bologna - Bologna, Italy

*Correspondence to:* Thomas Gastaldo (thomas.gastaldo2@unibo.it)

**Abstract.** Quantitative precipitation forecast (QPF) is still a challenge for numerical weather prediction (NWP), despite the continuous improvement of models and data assimilation systems. In this regard, the assimilation of radar reflectivity volumes should be beneficial, since the accuracy of analysis is the element that most affects short-term QPFs. Up to now, few attempts have been made to assimilate these observations in an operational set-up, due to the large amount of computational resources needed and to several open issues, like the arise of imbalances in the analyses and the estimation of the observational error. In this work, we evaluate the impact of the assimilation of radar reflectivity volumes employing a Local Ensemble Transform Kalman Filter (LETKF), implemented for the convection permitting model of the COnsortium for Small-scale MOdelling (COSMO). A 4 days test case on February 2017 is considered and the verification of QPFs is performed using the Fractions Skill Score (FSS) and the SAL technique, an object-based method which allows to decompose the error in precipitation fields in terms of structure ($S$), amplitude ($A$) and location ($L$). Results obtained assimilating both conventional data and radar reflectivity volumes are compared to those of the operational system of the Hydro-Meteo-Climate Service of the Emilia-Romagna region (Arpae-SIMC), in which only conventional observations are employed and latent heat nudging (LHN) is applied using surface rainfall intensity (SRI) estimated from the Italian radar network data. The impact of assimilating reflectivity volumes using LETKF in combination or not to LHN is assessed. Furthermore, some sensitivity tests are performed to evaluate the effects of the length of the assimilation window and of the reflectivity observational error (*roe*). Moreover, balance issues are assessed in terms of kinetic energy spectra and providing some examples of how these affect prognostic fields. Results show that the assimilation of reflectivity volumes has a positive impact on QPF accuracy in the first few hours of forecast both when it is combined to LHN or not. The improvement is further slightly enhanced when only observations collected close to the analysis time are assimilated, while the shortening of cycles length worsens QPF accuracy. Finally, the employment of a too small value of *roe* introduces imbalances in the analyses resulting in a severe degradation of forecast accuracy, especially when very short assimilation cycles are used.

## 1 Introduction

Numerical weather prediction (NWP) models are widely used in meteorological centres to produce forecasts of the state of the atmosphere. In particular, they play a key role in the forecast of precipitation (Cuo et al., 2011), which arouses a great interest due to the many applications in which it is involved, from the issue of severe weather warnings to decision making in several

branches of agriculture, industry and transportation. Therefore, an accurate quantitative precipitation forecast (QPF) is of great value for society and economic activities.

In recent years, the increase of available computing resources has allowed to increment NWP spatial resolution and to improve the accuracy of parametrization schemes, enabling to develop convection-permitting models (Clark et al., 2016). Despite that, QPF is still a challenge since it is affected by uncertainties in timing, location and intensity (Cuo et al., 2011; Röpnack et al., 2013). These errors arise partly from the chaotic behaviour of the atmosphere and from shortcomings in the model physics (Berner et al., 2015), but the main factor which affects the quality of QPF, especially in the short range (3-12 hours), is the accuracy of initial conditions (Dixon et al., 2009; Clark et al., 2016).

The initial condition (analysis) is generally produced by a data assimilation procedure which combines model state (background or first guess) and observations to provide the best possible estimate of the actual state of the atmosphere at a given time. In the last decades, different assimilation schemes have been proposed and implemented operationally in meteorological centres around the world (Bannister, 2016). They can be divided in different families: those based on a variational approach, like three-dimensional variational data assimilation (3D-Var: Courtier et al., 1998) and four-dimensional variational data assimilation (4D-Var: Buehner et al., 2010b), those based on the ensemble Kalman filter (EnKF: Evensen, 1994; Houtekamer and Mitchell, 1998) and those based on the particle filter (PF; see van Leeuwen, 2009 for a review). At the convective scale, EnKF methods seem to be preferable to variational schemes (Schraff et al., 2016). In fact, they determine explicitly the background error covariance, which is highly flow-dependent at the convective scale. Furthermore, in a variational scheme it is not straightforward to update any variable of a NWP model since an explicit linear and adjoint relation to the control vector of prognostic variables is needed. These problems can be partly addressed by employing hybrid EnKF-Variational techniques (like Wang et al., 2008; Gustafsson and Bojarova, 2014) but these approaches have mostly been applied to larger scale NWP. A more preferable option would be to employ a PF which is also considered to be the most promising technique to deal with the non-linear and non-Gaussian characters of dynamics and error statistics (Yano et al., 2018). Unfortunately, despite the efforts to overcome the dimensionality challenges of this assimilation technique (e.g. Poterjoy, 2016), PF is still not feasible for operational applications. Returning to EnKF methods, several variants have been suggested (for a survey refer to Meng and Zhang, 2011) and one of the most popular is the local ensemble transform Kalman filter (LETKF), proposed by Hunt et al. (2007). It is used operationally in several meteorological centres like at COMET (Bonavita et al., 2010), at MeteoSwiss employing the version of the scheme developed for the COSMO consortium (Schraff et al., 2016) and for research purposes at both the Japan Meteorological Agency (JMA; Miyoshi et al., 2010) and at the European Centre of Medium-Range Weather Forecasts (ECMWF; Hamrud et al., 2015).

The quality of the analysis is not determined only by the data assimilation scheme employed, but also by the quality and amount of observations that can be assimilated. With this aim, the assimilation of radar observations can be very beneficial, since they are highly dense in space (both horizontally and vertically) and in time. Up to now, several attempts have been made to improve the quality of analyses and subsequently the accuracy of QPFs by assimilating rainfall data estimated from radar reflectivity observations (Jones and Macpherson, 2006; Leuenberger and Rossa, 2007; Sokol, 2009; Davolio et al., 2017). Conversely, only few tries have been made to directly assimilate reflectivity volumes in a convection permitting model employing

EnKF techniques (e.g. Snyder and Zhang, 2003), especially in an operational framework (Bick et al., 2016). Despite some promising results, many issues affect the assimilation of reflectivity volumes at high spatial resolution and several aspects need to be further investigated.

First of all, the length of the assimilation window, which is one of the key aspects of any data assimilation system, has to be examined. In EnKF methods, a short window would be desirable to avoid that dynamical features leave the area where computed localized increments are significant (Buehner et al., 2010a) and to better preserve the gaussianity of the ensemble which can be compromised by non-linearities (Ferting et al., 2007). On the other hand, a too short window would lead to an increase of imbalances in the analysis, since the model has no the time to filter spurious gravity waves, introduced at each initialization, through the forecast step of the assimilation cycle. When reflectivity volumes are assimilated, the window length becomes even more crucial since these observations allow to catch small scale features of the atmosphere (Houtekamer and Zhang, 2016). In order to exploit the high temporal frequency of these data, which is essential to properly characterize fast developing and moving precipitation systems, it seems reasonable to employ short windows to assimilate, in each cycle, only observations collected very close to the analysis time. Furthermore, the choice of a short window is encouraged by the use of short localization scales, which has to be employed since small scales features are observed. Conversely, the big amount of radar observations enhances the imbalance issue and, therefore, the imbalances generated in the model by each initialisation should be checked and kept under control.

Another important issue is how to determine the observational error for radar reflectivities. As for any other observation, this is influenced by three different sources: instrumental errors, representativity errors and observation operator errors (Janjić et al., 2017). Since none of these are known, the choice of its value is not straightforward and can be estimated only in a statistical sense. Considering the amount of radar data, a correct estimation of the observational error is crucial, since even a small departure from the correct value can have a large impact on the quality of the analyses. Moreover, it should be taken into account that the use of the radar data is highly dependent on the observation operator adopted and its biases should also be studied and ideally removed. Finally, a further challenge is the estimation of the observational error correlation especially when dealing with radar data assimilation, due to the high density of this type of observations..

At Arpae-SIMC, the Hydro-Meteo-Climate Service of the Emilia-Romagna region, in Italy, a LETKF scheme is used to provide the initial conditions to the convection-permitting components of the operational modeling chain, consisting of one deterministic run and of one ensemble system both at 2.2 km of horizontal resolution. Currently, only conventional data are assimilated through the LETKF scheme and latent heat nudging (LHN; Stephan et al., 2008) is performed using rainfall intensity estimated from the Italian radar network data. The purpose of this paper is to present the first results obtained when also reflectivity volumes are assimilated using the LETKF scheme. In particular, the impact of assimilating reflectivity volumes in combination or not with LHN is evaluated. Furthermore, it is studied the sensitivity of the obtained analysis to two important characteristics of the assimilation cycle: the length of each cycle and the observational error attributed to the radar reflectivities.

This paper is organised as follows. In section 2 the model and the data assimilation system are described, as well as the observations employed. Furthermore, the operational set-up implemented at Arpae is reported in conjunction with the set-up

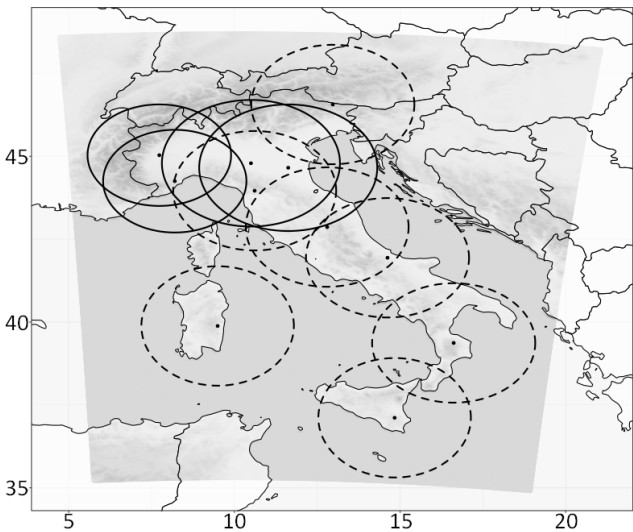

**Figure 1.** Integration domain (grayscale) of the COSMO model employed in this study with the Italian radar network overlapped. For each radar the approximate coverage area is shown with a dashed line if the radar system contributes only to the SRI composite employed in LHN and with a solid line if it is used also to directly assimilate reflectivity volumes through KENDA.

of the experiments performed in this study. In section 3 the verification methods are explained. In section 4 results are shown and discussed. In section 5 some conclusions are drawn.

## 2 Data, model and methodology

### 2.1 The COSMO model

The COSMO model (Baldauf et al., 2011) is a non-hydrostatic limited-area model developed by the multi-national COnsortium for Small-scale MOdelling (COSMO) and it is designed for both operational NWP and several research applications. It is based on the primitive equations describing compressible flows in a moist atmosphere and the continuity equation is replaced by a prognostic equation for the pressure perturbation (deviation from a reference state). The prognostic variables involved in these equations are the three dimensional wind vector, temperature, pressure perturbation, turbulent kinetic energy (TKE) and specific
amount of water vapour, cloud water, cloud ice, rain, snow and graupel.

In the present study, the COSMO model is run at 2.2 km horizontal resolution over a domain covering Italy and part of the neighbouring countries (Figure 1) and employing 65 terrain-following hybrid layers. The model top is at 22 km.

Regarding set-up and parametrizations, deep convection is resolved explicitly while the shallow convection is parametrized following the non-precipitating part of Tiedtke scheme (Tiedtke, 1989). Cloud formation and decay is controlled by a Lin-type
one moment bulk microphysics scheme which includes all the prognostic microphysical species (Lin et al., 1983; Seifert and Beheng, 2001). The turbulent parametrization is based on a TKE equation with a closure at level 2.5, according to Raschendor-

fer (2001). Radiative effects are described by the $\delta$-two-stream radiation scheme of Ritter and Geleyn (1992) for short-wave and long-wave fluxes. Finally, the lower boundary conditions at the ground are provided by the multi-layer soil model TERRA (Doms et al., 2011).

## 2.2 The KENDA system

The KENDA system (Schraff et al., 2016) implements for the COSMO model the LETKF scheme described by Hunt et al. (2007). In this implementation, the method is fully four dimensional, that is all observations collected during the assimilation window contribute to determine the analysis and the related model equivalents are computed using the prognostic variables at the proper observation time. To avoid spurious long-distance correlations in the background error covariance matrix, analyses are performed independently for each model grid point taking into account only nearby observations (observation localization).

Observations are weighted according to their distance from the grid point considered using the Gaspari-Cohn correlation function (Gaspari and Cohn, 1999). In the present work, two different values of the Gaspari-Cohn localization length-scale are employed for conventional and radar observations: 80 km for the former, 16 km for the latter (as done by Bick et al., 2016).

The limited size of the ensemble, combined to the assumption of a perfect model made in the LETKF scheme, leads to an underestimation of the background and analysis variances (e.g. Anderson, 2009) and, as a consequence, the quality of analyses

is negatively affected. To address this issue, KENDA provides some techniques to enlarge the spread of the ensemble (for a complete description of each of them refer to Schraff et al., 2016). Here, multiplicative covariance inflation (Anderson and Anderson, 1999) and the relaxation to prior perturbation (RTPP; Zhang et al., 2004) are employed. The former consists in inflating the analysis error covariance by a factor $\rho$ greater than one which is estimated following Houtekamer et al. (2005). The latter lies on the relaxation of the analysis ensemble perturbations $\mathbf{x}_i^a - \bar{\mathbf{x}}^a$ (where $\mathbf{x}_i^a$ is the analysis for the $i$-th member

and $\bar{\mathbf{x}}^a$ is the analysis ensemble mean) towards the background ensemble perturbations $\mathbf{x}_i^b - \bar{\mathbf{x}}^b$, that is:

$$\mathbf{x}_{i,new}^a - \bar{\mathbf{x}}^a = (1 - \alpha)(\mathbf{x}_i^a - \bar{\mathbf{x}}^a) + \alpha(\mathbf{x}_i^b - \bar{\mathbf{x}}^b) \tag{1}$$

where $\alpha_p = 0.75$ (see also Harnisch and Keil, 2015). Another approach provided by KENDA to account for model error is the additive inflation. It consists in adding random noise with mean $\mathbf{0}$ and covariance $\mathbf{Q}$ to the analysis ensemble members, where $\mathbf{Q}$ is the model error covariance matrix (Houtekamer and Mitchell, 2005). Since $\mathbf{Q}$ is not known, it is assumed to be

proportional (by a factor smaller than 1) to a static background error covariance $\mathbf{B}$ (Mitchell and Houtekamer, 2000). In the present work, additive inflation is not employed.

The KENDA suite also allows to compute the analysis weights, i.e. the analysis on ensemble space, on a coarsened grid (Yang et al., 2009). After being computed on the coarsened grid, weights are interpolated on the original high resolution grid and then used to compute analysis increments in model space. In this way, the computational cost is decreased without significantly

affecting the accuracy of analysis (Yang et al., 2009). In the present study, a coarsening factor equal to 3 is employed.

## 2.3 Assimilated data

KENDA allows the assimilation of both conventional and non conventional observations.

Conventional observations assimilated in this work include aircraft measurements (AMDAR) of temperature and horizontal wind, surface station measurements (SYNOP) of 10 m horizontal wind, 2 m temperature, 2 m relative humidity and surface pressure, radiosonde data (TEMP) of temperature, horizontal wind and humidity.

With regards to non conventional observation, KENDA allows also the assimilation of radar reflectivity volumes and radial winds. Radar data are assimilated through the Efficient Modular VOlume RADar Operator (EMVORADO) expressly designed for the COSMO model. It simulates the radar reflectivity factor and radial velocities processing the COSMO model fields one radar system at a time. Operator characteristics, resolution and the management of no-precipitation information are described in (Bick et al., 2016).

Although the operator gives the possibility to assimilate both radial winds and reflectivities, in the present work only reflectivity volumes are assimilated. Reflectivity volumes come from four different radar stations over Northern Italy (solid circles in Figure 1): Bric Della Croce (Piedmont Region), Settepani (Liguria Region), Gattatico and San Pietro Capofiume (Emilia-Romagna Region). Due to the complex orography of the considered area, radar are placed at very different altitudes and have different acquisition strategies. Observations are acquired every 10 minutes for Bric Della Croce radar, every 5 minutes for Settepani radar, every 15 minutes for San Pietro Capofiume radar and every 15 minutes starting from minutes 5 and 10 of each hour for Gattatico radar.

Data have a range resolution of 1 km, while the azimuthal resolution is 1 degree for Bric Della Croce and Settepani and 0.9 degree for San Pietro Capofiume and Gattatico. Before assimilation raw reflectivity are pre-processed taking into account non meteorological echoes, beam blocking and attenuation to improve the quality of data. In particular, it is important to eliminate the clutter signal that would affect the analysis retrieval introducing spurious observations. However, due to the fact that volumes from single radars undergo different pre-processing, it is not possible to define a homogeneous quality criterion. For this reason, all data in the volume that are not rejected from pre-processing step are supposed to have the same quality and are used into the assimilation cycle.

The high temporal and spatial density of observations is precious to estimate the initial state of numerical weather forecast. This allows to gather a lot of information on the real state of the atmosphere, but it determines an increase in analysis computational cost, in data transfer time and in memory disk occupation. Moreover, a spatial and/or temporal high density violates the assumption made in many data assimilation schemes: the non-correlation of observational errors. To reduce the total amount of data and to extract essential content of information, the superobbing technique is chosen (Michelson, 2003). In this way, reflectivities over a defined area are combined through a weighted mean into one single observation representative of the desired greater spatial scale. As in Bick et al. (2016), the horizontal resolution chosen in this work for the superobbing is equal to 10 km. Furthermore, before performing superobbing on the observed and simulated fields, a threshold of 5 dBZ is applied to both fields in order to avoid that large innovations associated to non-precipitating signals would lead to large analysis increments without physical relevance.

To evaluate the observational error associated to reflectivity volumes, a diagnostic based on statistical averages of observations-minus-background and observations-minus-analysis residuals, as described in Desroziers et al. (2005), is used. Employing all

radar data available during the test case, a reflectivity observational error (*roe*) equal to 5 dBZ is estimated, as found also by Tong and Xue (2005).

Finally, fields of surface rainfall intensity (SRI) are also assimilated in each member of the assimilation ensemble using a latent heat nudging scheme. SRI data come from the composite of the Italian radar network (all circles in Figure 1) and are distributed by the National Department of Civil Protection. These data have a temporal resolution of 10 minutes and a spatial resolution of 1 km, but before the assimilation they are interpolated at the model resolution. Data coming from each station undergo a quality control that removes those with low quality. The quality depends on different factors such as ground clutter, beam blocking, range distance, vertical variability and attenuation as described in Rinollo et al. (2013). The composite is then obtained as a weighted average of surface rain rates from single radar stations, where weights are represented by quality. These fields are assimilated through the LHN scheme, based on the assumption that the latent heat, integrated along the vertical column, is approximately proportional to the observed precipitation. The scheme, which is applied continuously during the integration of the model, acts in rescaling temperature profiles with an adjustment of the humidity field according to the ratio between observed and modelled rain rates. LHN has been gainfully employed in different frameworks, including forecasts over complex terrain (Leuenberger and Rossa, 2004; Leuenberger and Rossa, 2007). Our hypothesis is that, in the KENDA framework, LHN allows to have the model first guess closer to the observed atmospheric state, improving the analysis quality. For this reason, in all experiments (except one) presented here, LHN is applied together to the direct assimilation of reflectivity volumes through KENDA.

## 2.4 Operational set-up

The KENDA system is implemented operationally at Arpae using an ensemble of 20 members plus a deterministic run, which is obtained by applying the Kalman gain matrix for the ensemble mean to the innovations of the deterministic run itself. In principle, ensemble mean analyses can be deployed to initialize the deterministic forecasts, but this would lead to some inaccuracies since the mean of a non-Gaussian ensemble is generally not in balance (Schraff et al., 2016). For this reason the deterministic branch is added to the system, which differs from the ensemble ones only due to boundary conditions. The ensemble members use lateral boundary conditions provided every 3 hours at a 10 km horizontal resolution by the ensemble of the data assimilation system of the Centro Operativo per la Meteorologia (COMet), based on a LETKF scheme (Bonavita et al., 2010). The deterministic run employs hourly boundary conditions provided by a 5 km version of COSMO run at Arpae (COSMO-5M) which domain covers a large part of the Mediterranean basin and surrounding countries.

At Arpae, in the operational set-up, the COSMO model configuration described in Section 2.1 is adopted for all the 21 members. At present, in the operational chain only conventional observations are assimilated and LHN is performed on each member of the ensemble. The KENDA analyses are used operationally to provide initial conditions to COSMO-2I, the 2.2 km deterministic run initialized twice a day at 00 UTC and 12 UTC and to COSMO-2I EPS, an ensemble which is run every day at 00 UTC for a 48 hours forecast range.

## 2.5 Experimental set-up

In order to evaluate the impact of the assimilation of reflectivity radar volumes, several experiments are performed. Each experiment has the same set-up of the operational chain described in Section 2.4 regarding the number of members of the ensemble, boundary conditions and the COSMO model configuration. Therefore, they differ only due to the assimilation set-up. The complete list is provided in Table 1.

In *conv60* and *conv60_nolhn* experiments only conventional observations are assimilated using KENDA through cycles of 60 minutes. Moreover, in the former, LHN is performed during the forecast step of each assimilation cycle, replicating completely the operational set-up described in Section 2.4. Experiments *rad60* and *rad60_nolhn* are the analogous of *conv60* and *conv60_nolhn* when also radar reflectivity measurements are assimilated through KENDA, using a reflectivity observation error (*roe*) of 5 dBZ. A comparison between *conv60* and *rad60* or between *conv60_nolhn* and *rad60_nolhn* allows an assessment of whether, under the same conditions, the assimilation of reflectivity observations improves the quality of analyses. Furthermore, comparing *rad60* to *rad60_nolhn* it is possible to evaluate if the assimilation of reflectivity volumes combined with the LHN provides better results than the assimilation of only radar volumes.

| Trial | Window length [min] | Assimilated obs. | roe [dBZ] | Note |
|---|---|---|---|---|
| conv60 | 60 | conv. | - | - |
| conv60_nolhn | 60 | conv. | - | No LHN |
| rad60 | 60 | conv. + radar | 5 | - |
| rad60_nolhn | 60 | conv. + radar | 5 | No LHN |
| rad30 | 30 | conv. + radar | 5 | - |
| rad15 | 15 | conv. + radar | 5 | - |
| rad60_lst15 | 60 | conv. + radar | 5 | Use obs. in the last 15 min. of the window |
| rad60_roe10 | 60 | conv. + radar | 10 | - |
| rad60_roe0.5 | 60 | conv. + radar | 0.5 | - |
| rad15_roe10 | 15 | conv. + radar | 10 | - |
| rad15_roe0.5 | 15 | conv. + radar | 0.5 | - |

**Table 1.** Experimental set-up of each experiment including the length of the assimilation cycles, the type of observations assimilated, the reflectivity observation error (*roe)* associated to radar data and any additional feature.

All the other experiments involve the assimilation of both conventional data and reflectivity volumes, in addition to LHN. In order to test the impact of assimilating only observations which are not too far from the analysis time, sensitivity experiments on the duration of the assimilation windows are performed. This is tested by comparing *rad60* to experiments *rad30* and *rad15* which differ from *rad60* only for the length of the assimilation window, equal to 30 and 15 minutes respectively. An alternative way to assimilate only the most relevant observations is to select in each cycle a subset of data including the closest to the

analysis time. In the experiment *rad60_lst15* an assimilation window of 60 minutes is employed but only the observations (both conventional and radar reflectivities) collected in the last 15 minutes of the cycle are taken into account.

Since the estimation of observation error is not straightforward and different techniques can be applied, it is worth to evaluate the sensitivity of the assimilation system to this parameter. In addition to the value of 5 dBZ employed in the previous experiments, two other values are selected: 10 dBZ or 0.5 dBZ. Both of them are tested employing a 60 minutes assimilation window (*rad60_roe10* and *rad60_roe0.5*) and using 15 minutes cycles (*rad15_roe10* and *rad15_roe0.5*).

The experiments described above are carried out over a period of almost 4 days from February 3rd at 06 UTC to February 7th at 00 UTC in 2017. During 3 and 4 February, middle tropospheric circulation over Northern and Central Italy was dominated by southwesterly divergent flows associated with the passage of some precipitating systems. On February 5 a trough moved from France to Italy and this caused the formation of new precipitating systems in Northern Italy. During February 6 the trough moved slowly from Central Italy to the southern part of the country and precipitation systems weaken gradually. For each experiment, analyses of the deterministic member are used to initialize forecasts up to 24 hours every 3 hours from February 3 at 12 UTC to February 6 at 06 UTC with a total of 22 forecasts.

## 3  Verification

The performance of the experiments described in the previous section is assessed by the verification of precipitation employing three methods: comparison of areal average precipitation, SAL technique and Fractions Skill Score (FSS). The first method is applied only to the precipitation during the assimilation procedure for the deterministic member of the first 4 experiments in Table 1, while SAL and FSS are used to evaluate the QPF accuracy of the 22 forecasts initialized for each experiment.

### 3.1  Areal average precipitation

The method consists in comparing spatially averaged model precipitation to the average precipitation observed by rain-gauges over the same area. In order to have comparable samples, model precipitation is first interpolated on station locations by selecting the value at the nearest grid point. The rain-gauge stations employed for this verification method (nearly 1500) are the black dots in the dark grey region in Figure 2. This area is chosen to cover approximately the domain where reflectivity volumes are assimilated. Both model and rain-gauges precipitation are accumulated in 3 hours steps. Since this method is used for the verification during the assimilation procedure and the duration of each assimilation cycle, for the experiments considered, is 1 hour, model hourly precipitation is accumulated in order to obtain the 3-hourly precipitation. To summarize the results, the correspondence between model and observations is evaluated in terms of root mean square error (RMSE).

### 3.2  SAL

The SAL metrics (Wernli et al., 2008) is an object based verification score which allows to overcome the limitations of traditional scores for convection-permitting models, like the double-penalty problem (Rossa et al., 2008). The detection of individual objects in the accumulated precipitation fields is achieved by considering continuous areas of grid points exceeding a selected

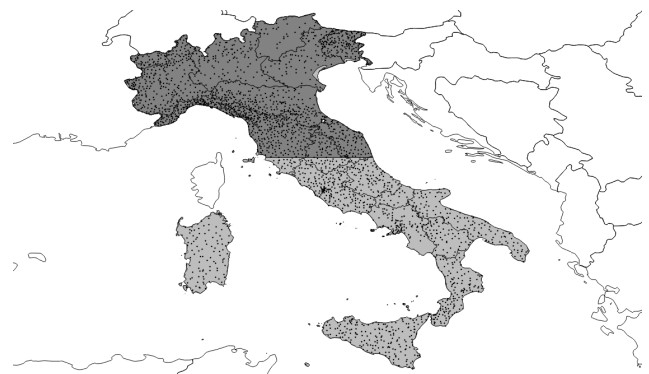

**Figure 2.** Verification domains employed to perform SAL (dark grey area) and FSS (union of dark grey and light grey areas). The rain-gauges (black dots) are used to correct precipitation estimated from the Italian radar network. Furthermore, rain-gauges over the black grey domain are employed for the verification of areal average precipitation during the assimilation procedure.

threshold. Comparing objects from observed and forecast fields, SAL provides information about the structure $S$, the amplitude $A$ and the location $L$ errors of QPF. A perfect match between forecast and observations would lead to $S = A = L = 0$; the more the values differ from 0, the greater the disagreement between model and observations. More in detail, a too sharp/flat (broad/small) structure of forecast precipitation compared to observations is associated to positive (negative) values of $S$; an
overestimation (underestimation) of average rainfall over the domain is associated to positive (negative) values of $A$; a misplacement of precipitating systems leads to positive values of $L$. Note that $L$ can range between 0 and 2, while $S$ and $A$ between -2 and 2.

Observations employed to perform SAL are hourly accumulated precipitation estimated from the Italian radar network and corrected using rain-gauges data. The radar-raingauges adjustment, adapted for a radar composite, derives from the method
described in Koinstinen and Puhakka (1981). The original method comprises two terms: a range dependency adjustment and a spatial varying adjustment. In our case, only the second term is taken into account due to the fact that, in overlapping areas of the composite, rainfall estimation is obtained combining data from different radars and, therefore, the original information on the range distance from the radar is lost. The correction is based on a weighted mean of the ratio between rain gauges and estimated radar rainfall amount calculated over the station locations. Weights are a function of the distance of the grid point
from the station and of a filtering parameter calculated as the mean spacing between 5 observations. Then a smoothing factor is applied to the correction.

The verification area, shown in dark gray in Figure 2, is the same as for the areal average precipitation. In this case, the rain-gauges inside it are employed to correct the rainfall estimation from the radar network. As mentioned before, this area covers approximately the domain where reflectivity volumes are assimilated. The choice of a larger domain would not be
feasible. In Wernli et al. (2009) it is recommended to use an area not larger than $500 \times 500$ km$^2$ since, otherwise, the domain may include different meteorological systems making the interpretation of results problematic. In fact, if the domain contains

strongly differing meteorological systems, then results obtained using the SAL technique may not be representative of the weakest one.

## 3.3 FSS

The Fractions Skill Score is a verification method introduced by Roberts and Lean (2008) based on the neighbourhood approach and applied to fractional coverage, that is the fraction of grid points exceeding a threshold. The score consists in comparing forecast and observed fractional coverage over squared box (neighbourhoods) and it ranges between 0 (completely wrong forecast) and 1 (perfect forecast). Therefore, a perfect match between model and observations is obtained when the two fields have the same frequency of events in each box. In this way, the method implicitly acknowledges that the actual resolution of a model is larger than the grid resolution and, at the same time, that also observations may contain random error at the model grid scale. Like SAL, this approach allows to overcome the limitation of traditional grid point based scores. Furthermore, it can be applied over a domain larger than that employed for SAL since it is based on dichotomy events instead of being based on the amount of precipitation. For this reason, in this work FSS is applied over the whole Italian country (union of dark gray and light gray domains in Figure 2) considering boxes of $0.2°$ in both latitude and longitude and, as for SAL, observations consist in hourly accumulated precipitation estimated from the Italian radar network corrected using rain-gauges data (all black dots in Figure 2).

## 4 Results

### 4.1 Impact of assimilating the radar reflectivities

A preliminary assessment of the impact of assimilating radar reflectivity volumes with the KENDA system is provided by comparing two pairs of experiments: *conv60_nolhn* with *rad60_nolhn* and *conv60* with *rad60*. In the experiment named *conv60_nolhn* only conventional observations are assimilated while in the *rad60_nolhn* experiment both conventional and radar reflectivity volumes are employed. The same dichotomy is preserved in the second pair of experiments but, in this case, LHN of SRI data is performed additionally in both *conv60* and *rad60*.

The areal average of 3-hourly precipitation during the assimilation procedure is displayed in Figure 3, employing precipitation recorded by rain-gauges (black line) as independent reference observation. Comparing *rad60_nolhn* (solid orange line) to *conv60_nolhn* (dashed blue line), the correspondence between forecast and observed precipitation is improved when reflectivity volumes are assimilated in combination with conventional data through KENDA. In fact, the root mean square error (RMSE) is reduced from 0.38 mm of *conv60_nolhn* to 0.26 mm of *rad60_nolhn* experiment. The same conclusion holds when the assimilation through KENDA is combined to LHN: the RMSE is reduced from 0.37 mm of *conv60* (dashed red line) to 0.29 mm of *rad60* (solid green line; this colour will be used from here onwards to identify uniquely this experiment). Note that the LHN substantially unaffects the overall agreement between forecast and observed precipitation when it is combined to the assimilation of only conventional data (RMSE equal to 0.38 mm for *conv60_nolhn* experiment and to 0.37 mm for

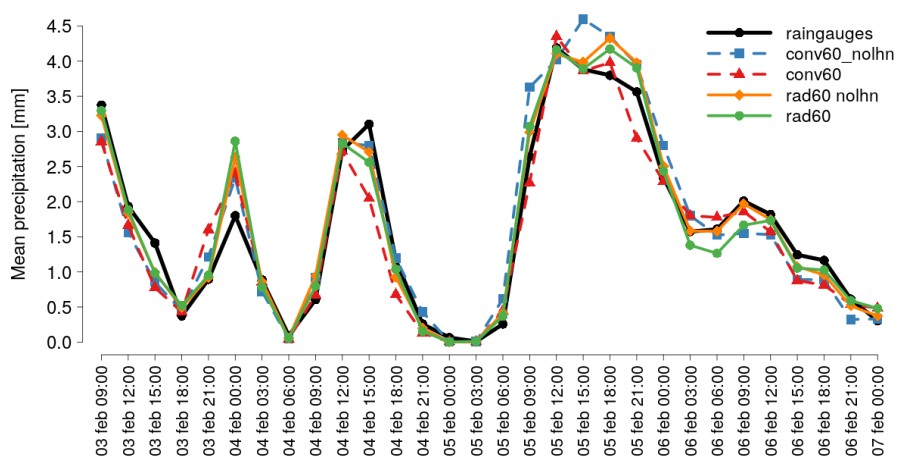

**Figure 3.** Areal average 3 hourly precipitation for rain-gauges (black) in the verification area shown in dark grey in Figure 2 and for the corresponding model forecast, during the assimilation procedure, relative to experiments *conv60_nolhn* (dashed blue line), *conv60* (dashed red line), *rad60_nolhn* (solid orange line) and *rad60* (solid green line).

*conv60*) while slightly degrades the correspondence when also reflectivity volumes are employed (RMSE equal to 0.26 mm for *rad60_nolhn* experiment and to 0.29 mm for *rad60*).

Verification of areal average precipitation during the assimilation procedure suggests that the quality of analyses is improved when radar reflectivity volumes are assimilated. To validate this result, the accuracy of QPF for the 22 forecasts initialized for
each experiment is evaluated. In order to give an insight about how analysis affects a forecast, hourly forecast precipitation from analyses on February 3 at 12 UTC are shown in Figure 4. Each column represents different lead times, from +1h to +3h going from left to right. The first row is the observed rainfall estimated from radars corrected by rain-gauges, that is the observed field employed for SAL and FSS described in Section 3. The shaded yellow area highlights the acquisition domain of the Italian radar network. The other rows are, in the order from top to bottom, the forecasts of the experiments *conv60_nolhn*,
*conv60*, *rad60_nolhn* and *rad60*. Forecast precipitation of *conv60_nolhn* is too weak and too spread, especially at lead time +2h in which large nuclei are forecast west of $12°$E. A significant improvement at +1h is obtained when considering *conv60*, even if a strong unobserved nucleus is forecast near $45.5°$N $13.5°$E, while at +2h and especially at +3h the precipitation is completely misplaced. When considering forecasts initialized from *rad60_nolhn* and *rad60*, rainfall accuracy at +1h is further enhanced in terms of location. Moreover, a significant improvement of both experiments compared to *conv60_nolhn* and *conv60* can be
noticed in location and intensity at lead times +2h and +3h. In particular, *rad60* is the only one able to forecast nuclei of the correct intensity with just a slight misplacement error.

For an objective verification of QPF, hourly precipitation of the 22 forecasts initialized from the analyses of each experiment is verified using SAL; to detect rainfall objects a 1 mm threshold is set. The verification using a threshold of 3 mm is also performed but, since results do not differ significantly from those obtained with a 1 mm threshold, they are not shown here.
Following the approach of Davolio et al. (2017), in Figure 5 the average of the absolute value of each component of SAL is

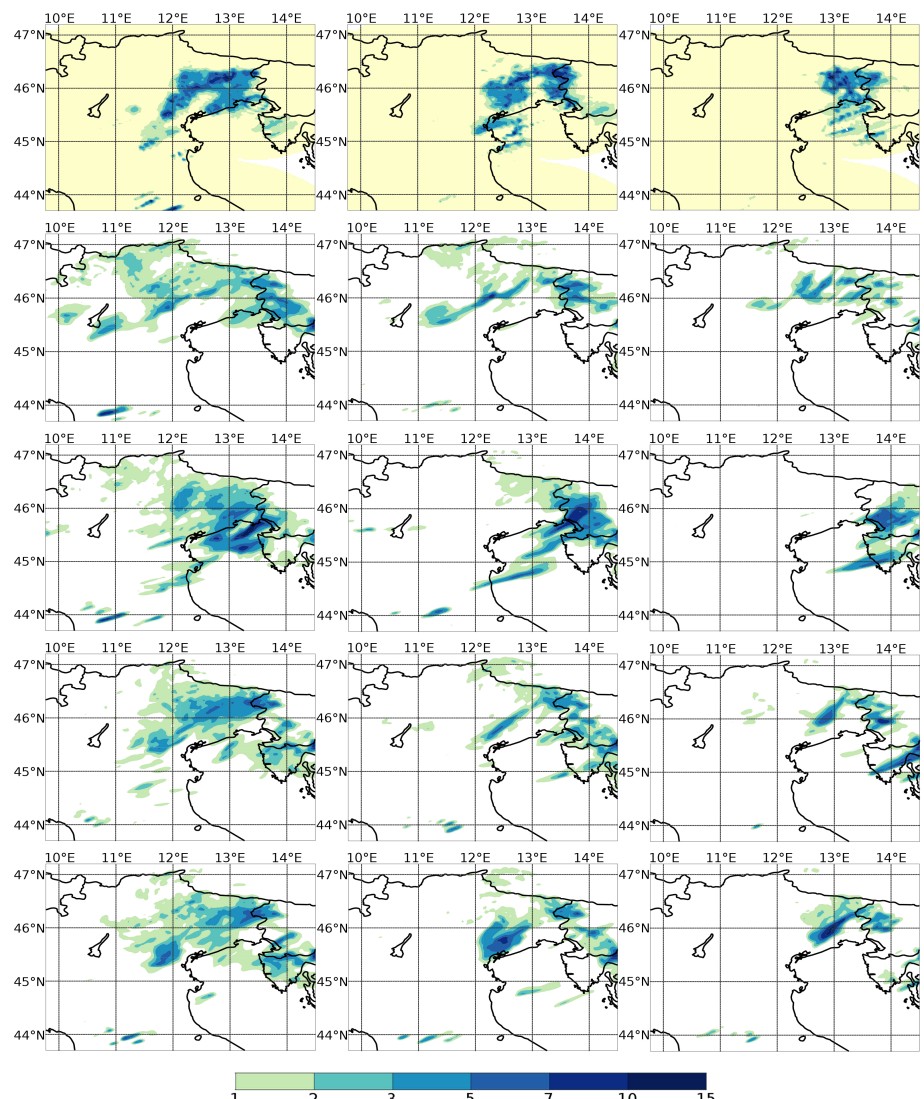

**Figure 4.** In the first row, the observed field, consisting in hourly rainfall estimated from radars corrected by rain-gauges, is shown for February 8 at 13, 14 and 15 UTC; the shaded yellow area highlights the acquisition domain of the Italian radar network. In the subsequent rows, forecast hourly precipitation of experiments *conv60_nolhn* (second row), *conv60* (third row), *rad60_nolhn* (fourth row) and *rad60* (fifth row) initialized on February 3 at 12 UTC is shown. Each column represents different lead times, from +1h to +3h going from left to right.

plotted as a function of lead time. Although forecasts are up to 24 hours, the verification is shown only for the first 8 hours, since after this lead time scores of the different experiments become very close. The average is computed considering only cases in which the observed or forecast rainfall field consists of at least 1000 grid points, which is approximately equal to an area of $50 \times 50$ km$^2$. Using the absolute value of the components of SAL, only the magnitude of the error is considered, loosing the information on the type of error (e.g., for $A$, an overestimation of forecast precipitation cannot be distinguished from an

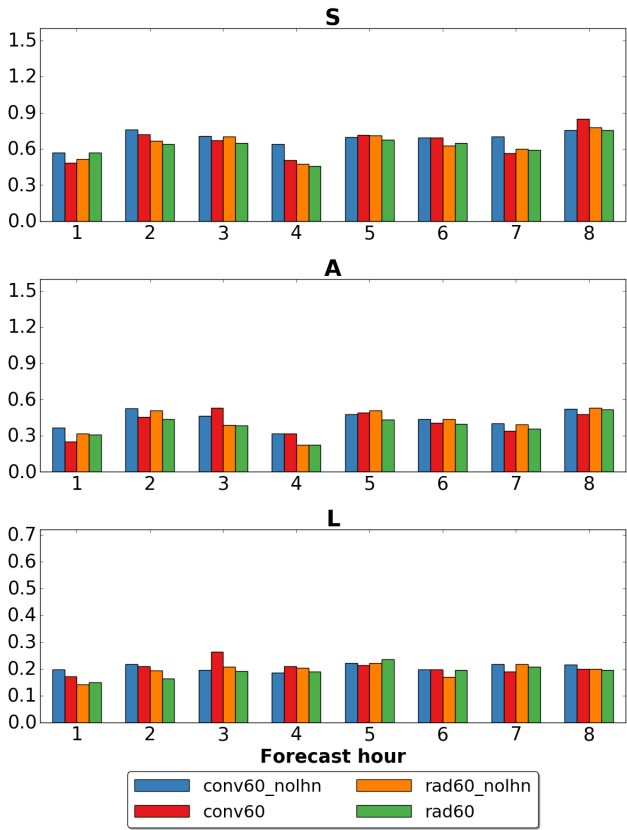

**Figure 5.** Average of the absolute value of each component of SAL over the 22 forecasts initialized from *conv60_nolhn* (blue), *conv60* (red), *rad60_nolhn* (orange) and *rad60* (green) analyses. Objects are selected using a threshold of 1 mm in hourly accumulated precipitation fields.. Cases in which the observed precipitation field consists of less than 1000 points are not taken into account in the average.

underestimation). This choice slightly limits the potential of SAL but provides an intuitive picture of the overall performance of each experiment.

Comparing *conv60_nolhn* to *rad60_nolhn*, QPF accuracy is slightly improved when forecasts are initialized from analyses obtained by assimilating both conventional data and reflectivity volumes instead of employing only conventional data. In fact,
5  at lead times +1h and +2h values of each component of SAL of *rad60_nolhn* are smaller than those of *conv60_nolhn*. An improvement can be noticed also at +3h in the *A* component, while *S* and *L* are substantially unaffected. At +4h *S* and *A* are improved while *L* is very slightly deteriorated. From +5h onwards, slight improvements and deteriorations alternate in an incoherent manner, therefore we can assess that the impact on QPF of using *conv60_nolhn* or *rad60_nolhn* analyses is substantially neutral.
10  The accuracy of QPF obtained by assimilating conventional data is improved at lead times +1h and +2h by activating LHN during the assimilation procedure: all components of SAL for *conv60* experiment are smaller than those of *conv60_nolhn*. The

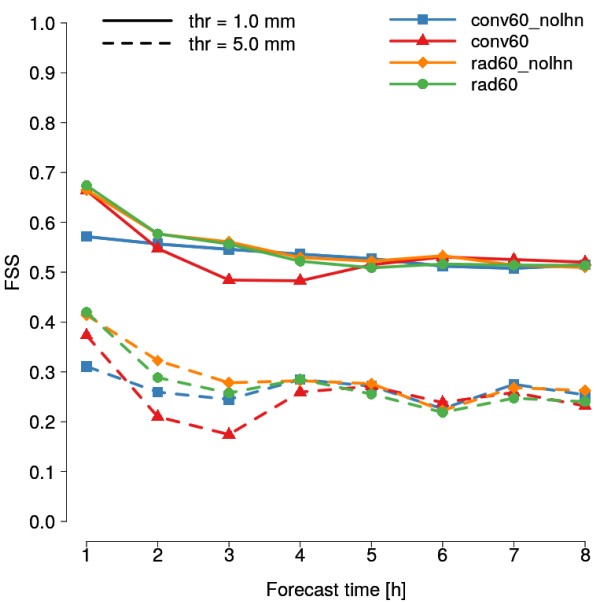

**Figure 6.** Fractions Skill Score as a function of lead time for *conv60_nolhn* (blue), *conv60* (red), *rad60_nolhn* (orange) and *rad60* (green). Verification is performed considering hourly precipitation and 1 mm (solid lines) and 5 mm (dashed lines) thresholds.

positive impact of LHN is already lost at +3h (in particular, a significant degradation in *L* component is observed) but, again, a benefit can be obtained by assimilating also reflectivity volumes. In fact, although at +1h the structure and amplitude errors of *rad60* are larger than those of *conv60* (while location error is slightly smaller), from +2h to +4h each component of SAL is smaller indicating a clear improvement in QPFs accuracy. Again, from +5h onwards, the impact of initializing forecasts from

different analyses becomes neutral. Regarding the combined use of LHN and the assimilation of reflectivity volumes through KENDA, at +1h the *S* component for *rad60* is slightly larger than that of *rad60_nolhn*, while *A* and *L* are almost equal. From +2h to +4h each component of SAL of *rad60* is always equal or slightly smaller than the corresponding one of *rad60_nolhn*.

Finally, to strengthen the results obtained using SAL over Northern Italy, the verification of QPF is extended to the whole Italian country employing FSS. Results are shown in Figure 6 for two thresholds: 1 mm (solid lines) and 5 mm (dashed lines).

Regarding the 1 mm threshold, a strong improvement in QPF accuracy can be noticed at +1h when reflectivity volumes are assimilated (*rad60_nolhn* and *rad60* experiments). At this lead time also only the use of LHN (*conv60*) is able to improve significantly the assimilation of conventional data (*conv60_nolhn*). At +2h the FSS value of the two experiments in which reflectivities are assimilated is still slightly larger than that of *conv60_nolhn*, while from +3h onwards differences become very small. Regarding *conv60*, the QPF accuracy strongly worsen between +2h and +4h and it is the worst among the 4 experiments.

Similar conclusions hold when the 5 mm threshold is considered but, in this case, QPF of experiments in which reflectivity volumes are assimilated outperforms *conv60* even at the first hour of forecast. Furthermore, values of FSS of *rad60_nolhn* are slightly larger than those of *rad60* between +2h and +4h.

In conclusion, summarizing the verification with SAL and FSS, at lead time +1h the assimilation of both conventional data and reflectivity volumes (*rad60* and *rad60_nolhn*) has a positive impact on QPF accuracy compared to the assimilation of only conventional data (*conv60_nolhn*). The improvement is seen not only where reflectivities are assimilated (Northern Italy), but also over a much larger area (the whole Italian country). Verification with SAL shows that a slight positive impact over Northern Italy holds up to +4h, while FSS scores reveal a benefit up to +2h over Italy for both 1 mm and 5 mm thresholds. The two experiments in which reflectivity volumes are assimilated do not substantially improve the QPF accuracy at +1h of the experiment in which only conventional data are assimilated in combination to LHN (*conv60*). However, they remarkably outperform it from +2h to +4h, as highlighted by both SAL and FSS. In this case, the positive impact is even enhanced when the 5 mm threshold is considered.

Finally, regarding the use of LHN combined to the assimilation of reflectivity volumes, SAL shows comparable results between *rad60* and *rad60_nolhn* at +1h, while QPF accuracy of *rad60* is very slightly enhanced compared to *rad60_nolhn* between +2h and +4h. Verification with FSS does not show significant differences between the two experiments for the 1 mm threshold, while *rad60_nolhn* very slightly outperforms *rad60* at +2h and +3h for the 5 mm threshold. Therefore, we can assess that QPF accuracy is substantially unaffected by assimilating twice an information derived from radar. On the basis of this result, even if we recognize that the combined assimilation of reflectivity volumes through KENDA and SRI by LHN may not be a rigorous process from a theoretical point of view, it is decided to keep the LHN for the subsequent experiments. In fact, this choice does not affect negatively the results of the sensitivity tests that are presented in this work and, at the same time, the LHN allows to use radar derived information on the state of the atmosphere in the whole Italian country, despite reflectivity volumes can be assimilated, at present, only over Northern Italy.

## 4.2   Impact of the length of the assimilation cycles

To obtain some insights about this topic, assimilation cycles of 15 and 30 minutes (respectively *rad15* and *rad30*) are tested and the results are compared to those obtained with the 60 minutes window (*rad60*), discussed in the previous subsection. Furthermore, an experiment in which observations are assimilated only if collected during the last 15 minutes of hourly assimilation cycles is performed (*rad60_lst15*). Accordingly, the total amount of assimilated data is reduced and the increments computed by the LETKF scheme should be more appropriate for computing the analysis, since the observations time is always very close to the analysis time.

In the same way as described in the previous subsection, QPF accuracy of the 22 forecasts initialized for each experiment is verified employing SAL and FSS. Results are shown, respectively, in Figure 7 and Figure 8 for *rad15* (red), *rad30* (orange), *rad60* (green) and *rad60_lst15* (blue). Considering SAL verification, at lead time +1h the shorter the cycle the smaller the error in structure and amplitude; however, the smallest location error among the experiments which differ only for the cycle length is associated to *rad30* while *rad15* and *rad60* are almost equal. Moreover, at lead time +1h also QPF of *rad60_lst15* is more accurate than that of *rad60* in each component. Between +2h and +4h, both *rad15* and *rad30* have always larger errors than *rad60*, with the only exception of *S* at +4h. In particular, a relevant worsening in the location of rainfall nuclei is observed at +3h. Regarding *rad60_lst15*, the comparison with *rad60* in the same forecast range reveals that the differences are always small

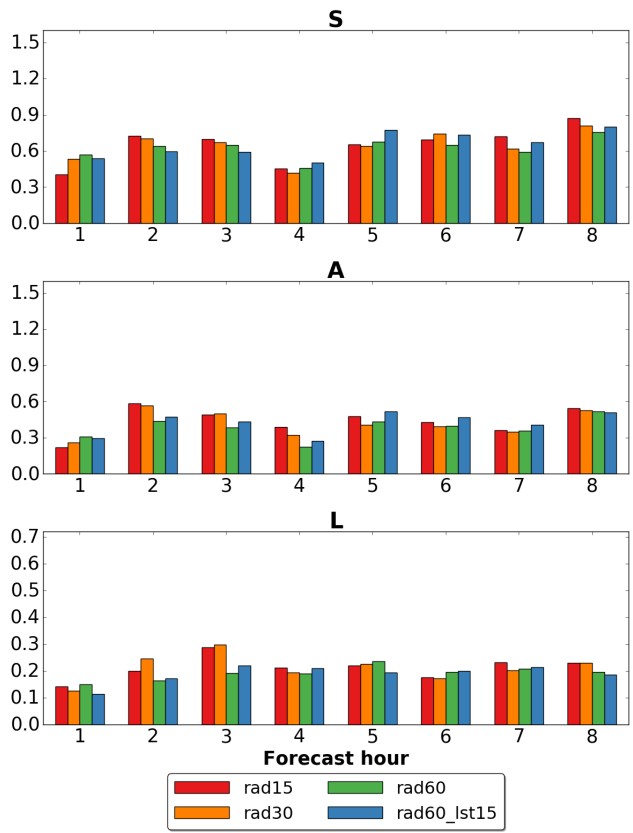

**Figure 7.** As in Figure 5 but considering experiments *rad15* (red), *rad30* (orange), *rad60* (green) and *rad60_lst15* (blue).

but, except for S at +2h and +3h, *rad60* slightly outperforms *rad60_lst15*. From lead time +5h onwards, differences among the 4 experiments become small and the results are mixed.

Extending the verification to the whole Italian country employing FSS, at +1h no significant differences can be noticed among the 4 experiments when the 1 mm threshold is considered. Between +2h and +4h, as observed with SAL verification, the shortening of the assimilation cycle worsens the QPF accuracy. Similarly, the differences between *rad60* and *rad60_lst15* are very small but, contrary to what observed with SAL, in this case the latter very slightly outperforms the former. From +5h onwards, FSS values of all the experiments are almost equal. When the 5 mm threshold is considered, the comparison between *rad15*, *rad30* and *rad60* leads to the same results as those observed for the 1 mm threshold with even more pronounced differences at lead times +2h and +3h. Regarding *rad60_lst15*, a significant improvement compared to *rad60* is noticed at +1h and a slight positive impact still holds at the subsequent lead times.

In summary, assimilating observations collected in the last 15 minutes of hourly cycles does not affect significantly the QPF accuracy when a 1 mm threshold is considered: at +1h a slight improvement is observed only over Northern Italy, while from +2h onwards the mixed results obtained with FSS and SAL suggest a neutral impact. However, *rad60_lst15* slightly

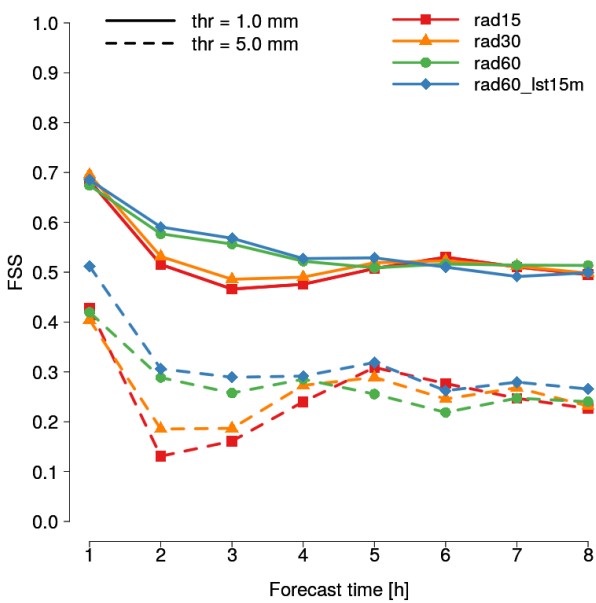

**Figure 8.** As in Figure 6 but considering experiments *rad15* (red), *rad30* (orange), *rad60* (green) and *rad60_lst15* (blue).

outperforms *rad60* over the whole Italian country when the 5 mm threshold is considered. Regarding the length of assimilation cycles, when it is shortened a slight improvement on QPF accuracy is observed at +1h over Northern Italy, but the impact over Italy is neutral. Thereafter, from +2h to +4h, a clear worsening is observed both where assimilation is performed and in the rest of Italy. To investigate if this worsening is due to the imbalance issue, the kinetic energy (KE) spectra of the

experiments is computed following the method described in Errico (1985). Curves displayed in Figure 9 are obtained as an average over the whole assimilation period (from 3 February at 06 UTC to 7 February at 00 UTC) of KE spectra computed each hour using analysis values of *u*, *v* and *w* over the whole domain. Kinetic energy spectra of *rad15* (red) and *rad60* (green) are almost overlapping, even at very small wavelength, indicating that shortening the length of cycles from 60 to 15 minutes does not introduce imbalances in the analyses (Skamarock, 2004). Furthermore, both spectra have a $-5/3$ dependence on the

wavenumber beyond a wavelength of 15-20 km, in agreement with observed spectra at the mesoscale (Nastrom and Gage, 1985). Same considerations apply also to KE spectra of *rad30* which is not shown. Therefore, with the current set-up, the use of a sub-hourly window length degrades QPF accuracy but this is not due to the introduction of imbalances in the analysis. A possible different explanation is that the reduced analysis error associated to *rad15* and *rad30* compared to *rad60* makes the ensemble employed for the LETKF scheme too small to correctly characterize the forecast error, as suggested in Uboldi and

Trevisan (2015).

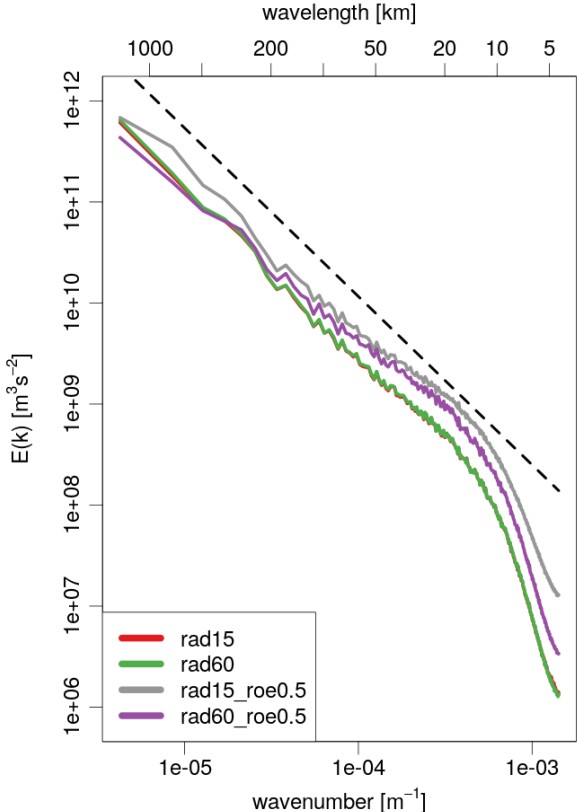

**Figure 9.** Kinetic energy (KE) spectra computed following the method described by Errico (1985). Each curve is obtained averaging KE spectra with a frequency of one hour during the assimilation procedure and employing analysis values of *u*, *v* and *w* over the whole model domain. The spectra are displayed for experiments *rad15* (red), *rad60* (green), *rad15_roe0.5* (grey) and *rad60_roe0.5* (violet). The dashed black line represents a function with a dependence to the wavenumber equal to $-5/3$.

## 4.3 Impact of changing the reflectivity observational error

A set of experiments is performed to investigate the impact of the reflectivity observational error in the assimilation scheme. In addition to the value of 5 dBZ employed so far, which was estimated applying the diagnostic described in Desroziers et al. (2005) to this case study, two other values of *roe* are tested: 10 dBZ and 0.5 dBZ. The former is employed by Bick et al. (2016) for the assimilation of reflectivity volumes from the German radar network using KENDA and COSMO and, therefore, should be reasonable also for the present study. The latter is a deliberately extreme value that may be chosen in the case of a great confidence in the quality of radar observations. These two different values of *roe* are used in assimilation cycles of 60 minutes

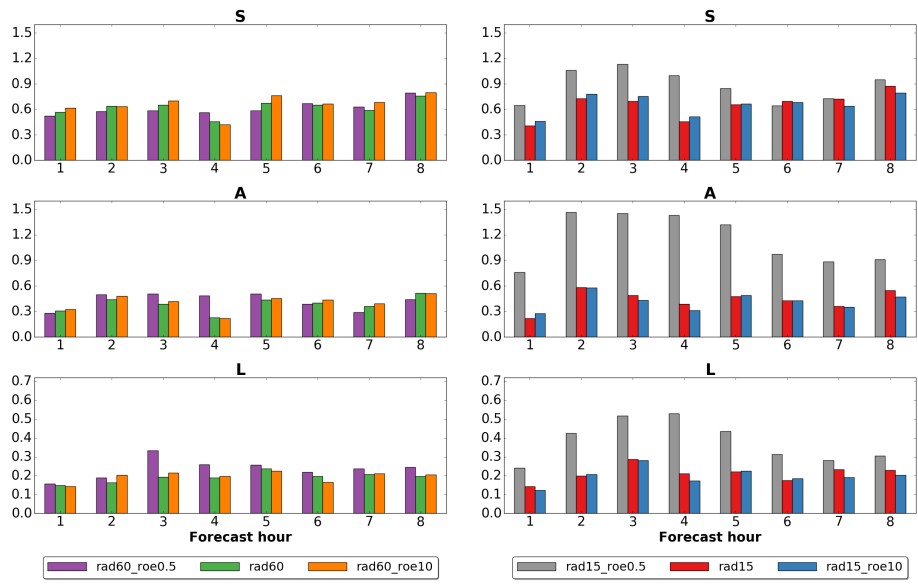

**Figure 10.** As in Figure 5 but considering, in the left panel, experiments *rad60_roe0.5* (violet), *rad60* (green) and *rad60_roe10* (orange) while, in the right panel, experiments *rad15_roe0.5* (grey), *rad15* (red) and *rad15_roe0.5* (blue).

(*rad60_roe0.5* and *rad60_roe10*) and 15 minutes (*rad15_roe0.5* and *rad15_roe10*). Therefore, they can be compared with the experiments with our standard value of roe = 5dBZ, respectively *rad60* and *rad15*.

Results of QPF verification in terms of SAL and FSS are reported, respectively, in Figure 10 and Figure 11. Regarding the experiments with a 60 minutes assimilation cycle, SAL verification (left panel) reveals that *rad60_roe0.5* slightly reduces
structure and amplitude errors on QPF at lead time +1h compared to *rad60*, but the location error is very slightly increased. From +2h to +4h, *rad60_roe0.5* has a larger error in all components, except *S* at +2h and +3h. In particular, the *A* component is remarkably worsened at +4h and the *L* component at +2h and +3h. As observed for the previous experiments, from +5h onwards the results become mixed. When comparing *rad60_roe10* to *rad60*, differences are small and mixed in the whole forecast range. The FSS verification carried out over the whole Italian country substantially confirms what observed with SAL:
*rad60_roe0.5* worsens QPF accuracy from +2h to +4h and the differences compared to *rad60* are even enhanced and extended to +1h when the 5 mm threshold is considered; at the same time, the impact of using a value of *roe* equal to 10 dBZ instead of 5 dBZ as a neutral impact over the whole forecast range.

In regards with 15 minutes assimilation cycles, *rad15_roe0.5* dramatically worsens QPF accuracy over Northern Italy in terms of structure (right panel in Figure 10) up to +5h and up to +12h in terms of amplitude and location (the range between
+9h and +12h is not shown). In this regard, the verification of individual forecasts (not shown here) reveals that the large error in *A* component is due to a systematic underestimation of the average precipitation over the domain. This marked worsening can be appreciated also with FSS verification (right panel in Figure 11), especially for the 1 mm threshold. When comparing results of SAL verification for *rad15_roe10* and *rad15*, differences are small and mixed over the whole forecast range. However, in this

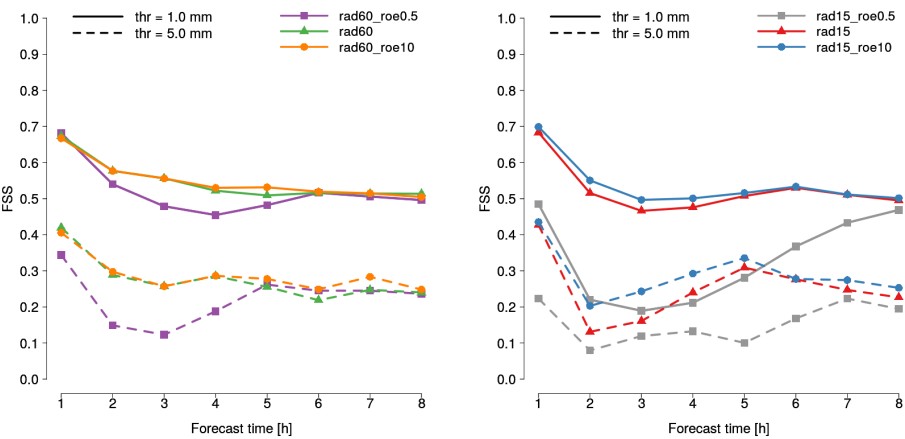

**Figure 11.** As in Figure 6 but considering, in the left panel, experiments *rad60_roe0.5* (violet), *rad60* (green) and *rad60_roe10* (orange) while, in the right panel, experiments *rad15_roe0.5* (grey), *rad15* (red) and *rad15_roe0.5* (blue).

case, FSS reveals that the former slightly outperforms the latter between +2h and +4h when the 1 mm threshold is considered and this is enhanced when considering the 5 mm threshold.

The overall poor quality of *rad15_roe0.5* forecasts is the direct consequence of the poor quality of the analyses from which they are initialized. As an example, in Figure 12 it is shown the mean sea level pressure (MSLP) and specific humidity at 850
5  hPa of *rad15_roe0.5* (right column) analysis on February 5 at 12 UTC and it is compared with the same quantities for the analysis of *rad15* (central column) and of the Integrated Forecasting System (IFS) of ECMWF (left column). Slight variations can be observed between IFS and *rad15* analyses and it seems reasonable that they may simply arise from differences between models and assimilation systems. Conversely, *rad15_roe0.5* analysis exhibits a noticeable increase in MSLP and a decrease in specific humidity over Northern Italy. This is in agreement with the decrease in forecast precipitation previously described.
10  In the same way as described in Section 4.2, KE spectra are computed for *rad15_roe0.5* and *rad60_roe0.5* and displayed in Figure 9. In both cases, at the smallest wavelength the KE is significantly greater that that of *rad15* or *rad60* and this is particularly evident for *rad15_roe0.5*. This behaviour is indicative of the presence of some undesired noise at small scales (Skamarock, 2004). Therefore, employing a value of *roe* equal to 0.5 dBZ, the assimilation system is not able to correctly remove small scale noise, especially when really short cycles are employed. Furthermore, the excess of energy associated to
15  the highest wavenumber modes propagates to the larger scales and the slope of the curves at wavelengths greater than 15 km differs from -5/3.

## 5   Conclusions

The assimilation of radar data in an operational set-up is a challenging issue. Most of the previous studies is devoted to the assimilation of rainfall estimated from radar data and it is currently widely employed in meteorological centres all over the

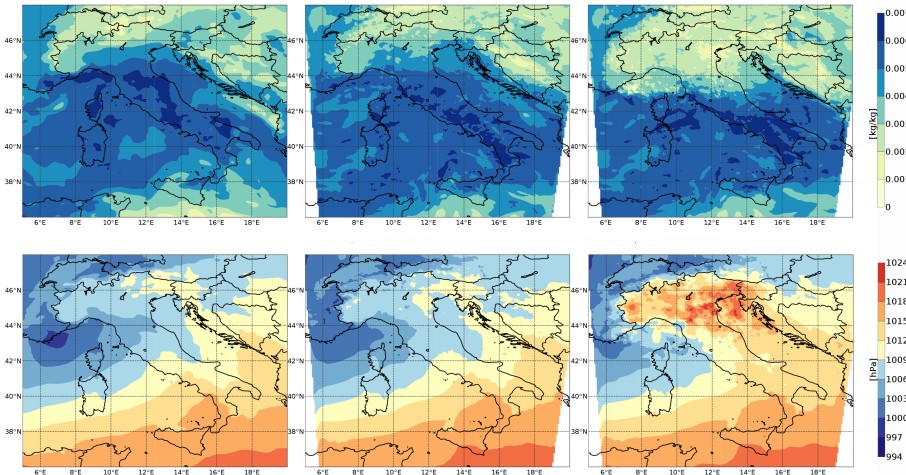

**Figure 12.** Mean sea level pressure (top) and specific humidity at 850 hPa (bottom) analysis on February 5 at 12 UTC for IFS (left) *rad60* (middle) and *rad15_roe0.5* (right).

world. The continuous increase of computer resources now allows to directly assimilate reflectivity volumes, but few studies have been dedicated to test the assimilation of these observations in an operational context. In the present work, the assimilation of reflectivity volumes using the LETKF scheme developed for the high resolution COSMO model is evaluated. A case study of 4 days in February 2017 is carried out using data from 4 radars over Northern Italy. The quality of the analyses generated

by the data assimilation system is assessed in terms of the accuracy of QPF which is verified using SAL (approximately in the region where reflectivity volumes are assimilated) and FSS (over the whole Italian country).

The assimilation of both conventional data and radar reflectivity volumes in combination to LHN (*rad60*) improves QPF accuracy compared to our operational set-up (*conv60*), in which only conventional data are employed together with LHN, and to *conv60_nolhn*, in which conventional observations are assimilated without performing LHN. The improvement compared to

*conv60* is remarkable between lead times +2h and +4h and observed both with SAL and FSS. Regarding the comparison with *conv60_nolhn*, the improvement is consistent at +1h and holds (attenuated) up to +4h over Northern Italy, while it becomes irrelevant over Italy from lead times +3h. Similar improvements are observed when both conventional data and reflectivity volumes are assimilated without LHN (*rad60_nolhn*), suggesting that the combined use of radar volumes and SRI, not rigorous from a theoretical point of view, does not degrade the results. This can be due to the different nature of the observed values: in

case of reflectivity volumes the measure is direct while for SRI the field is indirectly calculated using an empiric relationship between reflectivity and rain rate. Furthermore, also the assimilation schemes differ dramatically.

In this context, the assimilation of observations collected only in the last 15 minutes of each assimilation cycles (*rad60_lst15*), further slightly enhances the positive impact of assimilating reflectivity volumes. This result is observed when considering precipitation more intense than 5 mm/h while the impact of using *rad60_lst15* instead of *rad60* analyses is neutral when a 1 mm

threshold is employed. Taking into account that this configuration also reduces the computational cost associated to the assim-

ilation of radar data, it seems to be the most promising for an operational implementation. However, further tests would be necessary to evaluate if the same conclusion arises when only observations at the analysis time are assimilated. Regarding the length of assimilation cycles, the shortening of their length to 30 and 15 minutes slightly improves QPF accuracy at lead time +1h over the region where they are assimilated, but worsens results between lead times +2h and +4h both over Northern Italy and in the rest of the country. This is not due to the introduction of imbalances in the analyses. A possible explanation, which needs further investigation, is that the more frequent assimilation reduces the analysis error potentially making the ensemble spread too small to properly characterize the forecast error (Uboldi and Trevisan, 2015).

With regards to the observational error, it is found that a value of *roe* equal to 0.5 dBZ negatively affects the quality of the analyses and of the subsequent forecasts, because the model is not able to remove noise at the smallest scales. This leads to large errors in all prognostic fields in the area where radar data are assimilated and, as a consequence, to a very poor quality of the forecasts. This is particularly significant when 15 minutes assimilation cycles are employed, in which case forecast precipitation is strongly underestimated and misplaced. Conversely, a value of 10 dBZ, that is a value which is twice that estimated using Desroziers statistics, lead to similar results obtained with *roe* = 5 dBZ but slightly improves QPF accuracy when 15 minutes cycles are employed.

The observed improvement on QPF accuracy associated to the assimilation of reflectivity volumes is promising, even if it is limited only to the first few hours of forecast. Other tests are necessary to validate if this improvement holds in other synoptic conditions and for longer case studies. Furthermore, several tests need to be performed to extend the impact of the assimilation beyond the first few hours of forecast. In particular, the value of the reflectivity observational error seems to have a strong impact on QPF accuracy. Therefore, it seems reasonable that a further improvement can be achieved when *roe* is made dependent on the range, elevation, radar station and meteorological condition, but a better comprehension and estimation of this value is mandatory before testing more complex configurations.

*Competing interests.* No competing interests.

*Acknowledgements.* We would like to thank Roberto Cremonini for providing reflectivity volumes of the radars of Liguria and Piemonte regions and Anna Fornasiero for the correction of radar rainfall fields with rain-gauges. Finally, we thank the reviewers and the editor Alberto Carrassi for the numerous and valuable suggestions that helped us to improve the manuscript.

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
