# Peer review of "Data assimilation of radar reflectivity volumes in a LETKF scheme"

_Nonlinear Processes in Geophysics, 2018_

## Referee Comment (RC1) · Anonymous Referee #1 · 1 Jun 2018

In this paper several experiments are conducted to explore the sensitivity of a convective scale ensemble-based data assimilation system to the: length of the assimilation window, inclusion of additive inflation, variance of the reflectivity error. The experiments are conducted using real radar observations and combining latent heat nudging with the assimilation of conventional observations and radar reflectivity.

Although the topic is relevant and the experiments presented by the authors are interesting I think that some aspects of the manuscript needs improvement. I give details of these in the following comments.

Major comments:

Some aspects of the presentation needs to be improved. For example, some para-

graphs needs to be reorganized to improve the clarity of the manuscript. Also some figures can be merged in order to reduce the total number of figures in the paper. I provide examples of these changes as minor comments.

In this paper Latent Heat Nudging is combined with "direct" assimilation of radar reflectivity using EnKF. The introduction suggests that what is examined in this paper is the assimilation of radar reflectivity, however all the experiments use Latent Heat Nudging of precipitation estimated from radar reflectivity. I feel that the objective of the work should be reformulated since what is being explored is the added value of the LHN with "direct" assimilation of reflectivity. In this context an experiment which do not used LHN will also provide interesting results for comparison and discussion and will strength the conclusions. Also, if this is the focus of the paper references of previous work discussing these issues should be included in the introduction. I believe that LHN and EnKF has been combined in the development of the Rapid Update Cycle developed for the US.

Minor comments

P1L6 - We evaluated the impact or In this work the impact of . . . is evaluated P1L8 - SAL is not described. P1L9 - Missing stop before Results P1L12 of additive inflation P1L19 from the issue of . . . to decision making in . . . P1L22 - convection allowing models are a significant improvement in this direction. I think this should be mentioned explicitly because it allows the use of reflectivity data to improve the initial conditions. P2L9 - can you provide a reference for this? P2P14 - Recently particle filter has been successfully applied to convective allowing data assimilation see for example: Poterjoy, J., 2016: A Localized Particle Filter for High-Dimensional Nonlinear Systems. Mon. Wea. Rev., 144, 59–76, https://doi.org/10.1175/MWR-D-15-0163.1 P2L25 There are some previous work as well that deal with the issue of assimilation reflectivity in an EnKF starting (as far as I know) from the following paper: Snyder, C. and F. Zhang, 2003: Assimilation of Simulated Doppler Radar Observations with an Ensemble Kalman Filter. Mon. Wea. Rev., 131, 1663–1677, https://doi.org/10.1175//2555.1 P2L29 I think

this sentence may need more clarification. There is no mention to non-linear effects. I believe that one of the main reasons why a short window is desirable is because non linear effects will become important for longer windows. P2L32: Why when reflectivity volumes are assimilated the window length becomes more crucial? P3L1 by the use of short localization scales. P3L2 replace instability by imbalance (this also applies to other parts of the manuscript). P3L6 are known P3L4 In this paragraph the issue of observation error correlation should be mentioned as an additional challenge when dealing with radar data assimilation Figure 2, 3 and 4 can be merged into one single figure. P4L14 Is this scheme an online estimation scheme? What is the horizontal localization scale used in the experiments? P5L4 A discussion of possible implications of using a B matrix designed for low resolution models should be presented here. It would also be good to discuss previous work that shows positive impact associated with the inclusion of additive noise for convective scale data assimilation: Dowell, D.C. and L.J. Wicker, 2009: Additive Noise for Storm-Scale Ensemble Data Assimilation. J. Atmos. Oceanic Technol., 26, 911–927,https://doi.org/10.1175/2008JTECHA1156.1 P5L13, the paragraph starting here should be merged with the previous paragraph. P5L15 remove the :). Also it would be good to provide a reference for the quality control that is applied to radar data in general. How is the issues associated with complex terrain handled in this case (e.g. beam blocking) P6L1 This sentence is not clear please revise it. Figure 3 should include the effect of beam blocking to have a better idea of the area actually covered by the radar P6L15 Here it would be nice to add a reference for the data quality control. P7L5 Is the superobbing approach used also in the vertical? Is the superobbing considered in the observation operator as well? P7L10 What do the authors mean by "on average along the vertical"? P7L13 In this section a description of the experimental setting is presented. The clarity of the first two paragraphs needs to be improved. For example some operational systems are mentioned that are not used in the rest of the paper. It would be good to have a comparison between the operational systems and the experimental system, but in this section only the information regarding the experiments should be included. Table I caption, replace

trial by experiment Which is the output frequency of the model for the data assimilation cycle? P8L13 from February 3rd to February 7th P8L16 new precipitation systems Verification: It would be nice to show some examples of how the analysis look like and how the forecast look like in comparison with the observations. This will help to have a general idea on how well the system is working and how accurate the forecasts are. The figure in which the areal averaged precipitation is shown is based on the use of dependent data for the verification of the assimilation system. The authors said that since all the experiment are verified in this way this should not be a problem. However for me this is not convincing, since validating with dependent data might not detect issues like overfitting. An analysis closes to the observation is not necessarily the best analysis. P9L15 Describe the SAL acronym P9L17 Here the authors said that SRI observations are not independent, however the SAL approach is applied to the precipitation forecasts and not to the analysis. In this sense the observations are independent because these observations has not been assimilated yet. P9L30 Please provide more detailed explanation on this limitation of SAL. My understanding from this paragraph is that SAL can only work with one precipitation system at a time. But this is difficult to guarantee even if the domain is very small as proposed by the authors. P10L16 Only results concerning 1 mm are shown. Are the results sensitive to the threshold used in the SAL method? Can the authors comment on the results obtained with other thresholds as well? P12L8 It is not clear how the kinetic energy spectra can be used to identify the effect of the imbalance. May be the evolution of the spectra with the forecast lead time would be a better tool to detect the presence of small scale noise that arises as a consequence of the assimilation of observations (in a similar way as it is done with pressure tendencies).

---

## Referee Comment (RC2) · Anonymous Referee #2 · 18 Jun 2018

Review of "Data assimilation of radar reflectivity volumes in a LETKF scheme " by Thomas Gastaldo et al.

General comments:

I do recognize the authors' efforts on assimilating the radar data in their regional data assimilation system, KENDA. However, I felt that the setups of the experiments cannot clearly illustrate the impact of radar reflectivity on precipitation prediction, given that radar information has been injected into the model state through latent heat nudging. Also, with a high-resolution setup, it is somewhat surprising that there is no benefit from more rapid updates. I am also concerned a potential systematic underestimation of precipitation (and humidity) when a strong dependence on radar data is tested. These seem to lead to issues of radar data quality or how the authors handle the raw radar data.
Based on these concerns, I will recommend major revision for this manuscript and hope the authors can address the following comments in their revised manuscript.

Major comments:
1. I understand that the assimilation configuration used in this study attempts to be close to the operation settings as much as they could. However. a big question is whether the justification of the impact from radar data on precipitation is fair, given that the precipitation nudging is always applied and the observations for verification contains both information of radar and surface rain gauges. Is it possible to conduct experiments without LHN for clean comparison? E.g. an experiment assimilates conventional data only and compares with the experiment that assimilates conventional and radar data. And, compare the results with the rain gauges data?
    o Does the result imply that LHN is more effective than radar data assimilation?
2. Intuitively, the assimilation of radar data is expected to improve the precipitation. It doesn't seem to be a reasonable choice to me that only examining the absolute value of the components of SAL, without trying to distinguish the possibility of overforecasting or underestimation of the precipitation.
    o In most of the literate using radar data, the impact is mostly seen in the first 6-h forecast and some even only last for 3 hours. Do the authors see a clear impact for the 1-h or 2-h lead time?
3. It is not too surprising to me that rad60_BM has a worse performance since the perturbations used to augment the ensemble-based background error covariance may be in larger scale. I will suggest either remove this experiment or illustrate the reasons that degrades the performance.
4. It is unclear to attribute the degradation of using a sub-hourly assimilation window to location of rainfall nuclei (Page 13, line, 4). Can the authors explain why a more rapid update (15 or 30-min window) lead to a worse performance than the one using a 60-min window since using a short assimilation window does not introduce the imbalance issue?
5. The authors explain that a larger A component in SAL with the use of small observation error (roe0.5) is due to a systematic underestimation of the average precipitation over the domain or as the example showing a result of decreased humidity. With a strong confidence in observations, such results will be highly

dominated by the characteristics of the radar reflectivity data. Do the authors observe that the rain estimated by radar data is underestimated as compared with the rain gauge data? Is there a calibration issue such as the attenuating effect in radar data or the QC procedure (O-B > 5dBz) before the superobservations were constructed?

- o In the experiments of rad60_roe10 and rad60_roe0.5, is the QC during assimilation the same?
- o I don't quite follow with the rationale in the last paragraph on Page 15. With rad15_roe0.5, It should be the assimilation introduces the small-scale features, instead of losing the ability to "correct" the small-scale errors. To verify this, I suggest that the authors can compare the KE spectrum before (background) and after (analysis) assimilation.

6. Information and setups about Radar data assimilation are not clear.
    - o Although Bick et al. (2016) had described the operator characteristics, and other radar data management. It will still be essential for this paper to briefly provide the important information related to data assimilation. For example, the volume used to construct the superobservation (degree, gate, etc..?). Are all the radar data from different observation intervals with different radars used for constructing the superobservations?
    - o Page 7, line 6: Is there a precondition to reject (O-B > 5dBz) to avoid large innovations associated to non-precipitating signals? If (O-B >5dBz), doesn't it mean that observation tend to have more rain water? Are the assimilation/forecast results sensitive to such choice?
    - o If the horizontal grid-spacing of the analysis domain is 2.2km, isn't it too coarse to have superobservations with resolution of 10km?
    - o Since Bick et al. (2016) used an observation error of 10dBz, is there a particular reason why this study reduces the observation error to 5dBz?
    - o Page 7, line 28: Isn't the radial velocity also assimilated? It's not clear to me why the authors only emphasize on the contribution from reflectivity.

**Minor comments**
Please provide the following Information

- Page 3, line 28: what is the model top of the model?
- Page 5, line 4: please spell out the full name of the ICON model.
- Page 5, line11, 14: It's not clear the composite map is composed by what data? Radar only? Or weighted average with the surface rain rate? Is this the same as the observations employed to perform SAL? (Page 9, lines 16-17)
- Page, 7, line19: I would suggest to cite the original reference for the LETKF scheme: Hunt et al. 2007.
- Should I assume that the horizontal grid-spacing of the domain for assimilation is 2.2km?
- Page 10, line 20: "observed rainfall field consists of at least 1000 grid points"=> It would be better to change the observed target based on the definition of area (e.g. 50km x 50km?).
- Page 16, line 3-4: Actually, a lot of efforts have been devoted to assimilation of radar reflectivity data already. I am not sure why the authors have such statement.

---

## Editor Comment (EC1) · A. Carrassi (Editor) · 18 Jun 2018

Dear Authors,

while both Reviewers agree on the relevance of your work, they also identified some weaknesses that must be addressed.

I strongly encouraged to do that and to submit a new version. In particular, and as suggested by both Reviewers, I invite you to consider running an experiment without LHN, whose presence renders impossible (or very difficult) to assess the impact of radar data.

I will be happy to receive a new version of your manuscript.

With best regards,

Alberto Carrassi

---

## Author Comment (AC1) · 21 Jun 2018

Dear Alberto Carrassi,

we will provide results for a run without LHN. Furthermore, the article will be integrated to respond to the comments of reviewrs.

Thanks to you and to the reviewers.

Regards

Thomas Gastaldo

---

## Editor Comment (EC2) · A. Carrassi (Editor) · 24 Jul 2018

Dear Authors,

thanks for submitting a revised version of your manuscript. While the new version is assessed by the Reviewers, I would like to add some minor suggestions/comments:

1) the authors might want to relate the general context of their study to a recent essay work: Yano et al., 2018: Scientific challenges of convective-scale numerical weather prediction. BAMS https://doi.org/10.1175/BAMS-D-17-0125.1.

2) I suggest the authors to include a brief mention of the conclusion they are drawing at the end of the abstract. In its present form it well describes the problem tackled and the methodology therein, but not much on the authors' main conclusion.

[Figure]

3) in relation with the somehow unexpected lack of improvement when the data assimilation interval is further reduced, the authors might want to relate/interpretate it based on the study on dynamical instabilities within a convective-resolving model by Uboldi and Trevisan, 2015: Multiple-scale error growth in a convection-resolving model. Nonlinear Processes in Geophysics 22, 1–13, 2015.

Best Regards,

Alberto

---

## Author Response (AR1)

**Referee #1**

In this paper several experiments are conducted to explore the sensitivity of a convective scale ensemble-based data assimilation system to the: length of the assimilation window, inclusion of additive inflation, variance of the reflectivity error. The experiments are conducted using real radar observations and combining latent heat nudging with the assimilation of conventional observations and radar reflectivity. Although the topic is relevant and the experiments presented by the authors are interesting. I think that some aspects of the manuscript needs improvement. I give details of these in the following comments.

**Major comments:**

Some aspects of the presentation needs to be improved. For example, some paragraphs needs to be reorganized to improve the clarity of the manuscript. Also some figures can be merged in order to reduce the total number of figures in the paper. I provide examples of these changes as minor comments.

A reply to each comment is provided in the "Minor comments" section, indicating also the changes made to the document.

In this paper Latent Heat Nudging is combined with "direct" assimilation of radar reflectivity using EnKF. The introduction suggests that what is examined in this paper is the assimilation of radar reflectivity, however all the experiments use Latent Heat Nudging of precipitation estimated from radar reflectivity. I feel that the objective of the work should be reformulated since what is being explored is the added value of the LHN with "direct" assimilation of reflectivity. In this context an experiment which do not used LHN will also provide interesting results for comparison and discussion and will strength the conclusions. Also, if this is the focus of the paper references of previous work discussing these issues should be included in the introduction. I believe that LHN and EnKF has been combined in the development of the Rapid Update Cycle developed for the US.

As suggested by both the reviewers, an experiment (rad60_nolhn) in which conventional data and reflectivity volumes are assimilated without LHN is added to the manuscript. Details of the set-up employed are described in section 2.4 (as for all the other experiments) while results and the comparison with rad60 (in which LHN is applied together to the assimilation of reflectivity volumes) are provided in section 3.1.

No significant changes can be noticed in the results when comparing rad60 and rad60_nolhn. Therefore, it is decided to not switch off the LHN for the other experiments. In fact, this choice does not affect the results of the sensitivity tests that are presented in this work and, at the same time, the LHN allows to use radar derived information on the state of the atmosphere in the whole Italian country, despite reflectivity volumes can be assimilated, at present, only over Northern Italy.

Some modifications and additions are made in the abstract, in the introduction and in the conclusions to better empathize the use of LHN in combination to the assimilation of reflectivity volumes. Furthermore, it is explicitied that a test is performed to evaluate the impact of switching off LHN when reflectivity volumes are assimilated

**Minor comments**

P1L6 -We evaluated the impact or In this work the impact of … is evaluated

Done.

P1L8 – SAL is not described.

A brief explanation of the SAL technique was already provided but, to improve the clarity, the sentence is modified as follows:

A 4 days test case on February 2017 is considered and the verification of QPFs is performed using the SAL technique, an object-based method which allows to decompose the error in precipitation fields in terms of structure (S), amplitude (A) and location (L).

P1L9 - Missing stop before Results

Done.

P1L12 of additive inflation
Done.

P1L19 from the issue of … to decision making in …
Done.

P1L22 - convection allowing models are a significant improvement in this direction. I think this should be mentioned explicitly because it allows the use of reflectivity data to improve the initial conditions.
The sentence is modified as follows:
In recent years, the increase of available computing resources has allowed to increment NWP spatial resolution and to improve the accuracy of parametrization schemes, enabling to develop convection-permitting models (Clark et al., 2016).

P2L9 can you provide a reference for this?
Done: Schraff et al., 2016.

P2P14 - Recently particle filter has been successfully applied to convective allowing data assimilation see for example: Poterjoy, J.,2016: A Localized Particle Filter for High-Dimensional Nonlinear Systems. Mon. Wea. Rev., 144, 59–76, https://doi.org/10.1175/MWR-D-15-0163.1
This sentence is added and a previous sentence (in the new version of the manuscript is P2L10) is changed according to this addition:
Another option may be to employ particle filters but, despite the efforts to overcome the dimensionality challenges of these assimilation techniques (e.g. Poterjoy, 2016), it is still not feasible for operational applications.

P2L25 There are some previous work as well that deal with the issue of assimilation reflectivity in an EnKF starting (as far as I know) from the following paper: Snyder, C. and F. Zhang, 2003: Assimilation of Simulated Doppler Radar Observations with an Ensemble Kalman Filter. Mon. Wea. Rev., 131, 1663–1677, https://doi.org/10.1175//2555.1
The sentence and the subsequent one are modified as follows:
Conversely, only few tries have been made to directly assimilate reflectivity volumes in a convection permitting model employing EnKF techniques (e.g. Snyder and Zhang, 2003), especially in an operational framework (Bick et al., 2016). Despite some promising results, many issues affect the assimilation of reflectivity volumes at high spatial resolution and several aspects need to be further investigated.

P2L29 I think this sentence may need more clarification. There is no mention to non-linear effects. I believe that one of the main reasons why a short window is desirable is because non linear effects will become important for longer windows.
The sentence is modified as follows:
In EnKF methods, a short window would be desirable to avoid that dynamical features leave the area where computed localized increments are significant (Buehner et al., 2010a) and to better preserve the gaussianity of the ensemble which can be compromised by non-linearities (Ferting et al., 2007).

P2L32: Why when reflectivity volumes are assimilated the window length becomes more crucial?
Due to their high resolution, reflectivity observations can better define the small-scale features of weather phenomena. To improve the understanding, the sentence is modified as follow:
When reflectivity volumes are assimilated, the window length becomes even more crucial since these observations allow to catch small scale features of the atmosphere (Houtekamer and Zhang, 2016).

P3L1 by the use of short localization scales.
Done.

P3L2 replace instability by imbalance (this also applies to other parts of the manuscript).
Done.

P3L6 are known
Done.

P3L4 In this paragraph the issue of observation error correlation should be mentioned as an additional challenge when dealing with radar data assimilation.
This sentence is added:
Finally, a further challenge is the estimation of the observational error correlation especially when dealing with radar data assimilation, due to the high density of this type of observations.

Figure 2, 3 and 4 can be merged into one single figure.
We merged figures 2 and 3. We have decided to keep Figure 4 separate as it relates purely to the verification of results. In addition, we decided to plot, over the verification domain, the rain-gauges stations used in the verification. As a result, some small modifications are made in sections 2.3 and 2.5.

P4L14 Is this scheme an online estimation scheme? What is the horizontal localization scale used in the experiments?
RTPP is an online scheme.
For horizontal localization, an 80km length scale is used for conventional data (as Bick, 2016) while 16km for radar volumes (as Bick, 2016). The following sentences are added at the first paragraph of section 2.2 to include these information:
To avoid spurious long-distance correlations in the background error covariance matrix, analyses are performed independently for each model grid point taking into account only nearby observations (observation localization). Observations are weighted according to their distance from the grid point considered using the Gaspari-Cohn correlation function (Gaspari and Cohn, 1999). In the present work, two different values of the Gaspari-Cohn localization length-scale are employed for conventional and radar observations: 80 km for the former, 16 km for the latter (as done by Bick et al., 2016).

P5L4 A discussion of possible implications of using a B matrix designed for low resolution models should be presented here. It would also be good to discuss previous work that shows positive impact associated with the inclusion of additive noise for convective scale data assimilation: Dowell, D.C. and L.J. Wicker, 2009: Additive Noise for Storm-Scale Ensemble Data Assimilation. J. Atmos. Oceanic Technol., 26, 911–927,https://doi.org/10.1175/2008JTECHA1156.1
Some additions and modifications have been made in the last part of the paragraph:
Since Q is not known, it is assumed to be proportional (by a factor smaller than 1) to a static background error covariance B (Mitchell and Houtekamer, 2000). This technique has already been employed with a positive impact in convective scale data assimilation (e.g. Dowell and Wicker, 2009). In the present work, additive inflation is used together with multiplicative inflation and to RTPP only in one experiment, employing a climatological B-matrix from the 3D-VAR of the Icosahedral Nonhydrostatic (ICON) global model (Zängl et al., 2015). Although the use of a lower resolution B-matrix may not allow to properly characterize the model error at the smallest scales, the same configuration has been gainfully employed at MeteoSwiss (Leuenberger and Merker, 2018).

P5L13, the paragraph starting here should be merged with the previous paragraph.
Done. Note that the whole paragraph regarding LHN has been moved at the end of the section. In this way, first all observations (conventional and non conventional) assimilated by KENDA are described, then those employed for the LHN are described.

P5L15 remove the :). Also it would be good to provide a reference for the quality control that is applied to radar data in general.

The reference for the quality control has been added and the statement has been changed to better explain how this control is carried out, as follows:

Data coming from each station undergo a quality control that removes those with low quality. The quality depends on different factors such as ground clutter, beam blocking, range distance, vertical variability and attenuation as described in Rinollo et al. (2013).

How is the issues associated with complex terrain handled in this case (e.g. beam blocking)

For the SRI product the effect of the beam blocking is combined with the other parameters that enter into the generation of quality. The use of different radars to generate the composite, taking for different radars the points with the highest quality, fulfill the domain and the direct beam blocking effect is lost.

P6L1 This sentence is not clear please revise it.

The sentence and the previous one are modified as follows:

The scheme, which is applied continuously during the integration of the model, acts in rescaling temperature profiles with an adjustment of the humidity field according to the ratio between observed and modelled rain rates. LHN has been gainfully employed in different frameworks, including forecasts over complex terrain (Leuenberger and Rossa, 2004; Leuenberger and Rossa, 2007).

Figure 3 should include the effect of beam blocking to have a better idea of the area actually covered by the radar

The figure 3 has been merged to figure 2.

As specified in two previous answers, due to the fact that it arises from a composite, the SRI product fulfill the domain presented in figure. The four radars highlighted in red are assimilated in the KENDA system taking all data volumes, for this reason it has less sense to indicate where radar beams are blocked, because this affects only first elevations and elevations higher than firsts have the same domain. The caption of the figure is misleading. For this reason it has been changed.

P6L15 Here it would be nice to add a reference for the data quality control.

Reflectivity volumes used in the KENDA systems come directly from two different Regional Meteorological Services. For this reason they are subject to different cleaning procedures described, if present, by internal documentation in italian, not suitable for being used as a reference.

P7L5 Is the superobbing approach used also in the vertical? Is the superobbing considered in the observation operator as well?

Superobbing is applied only in the horizontal to both the observed and simulated field, as well as the threshold of 5 dBZ.. To avoid misunderstandings about this, the subsequent sentence is modified as follows:

Furthermore, before performing superobbing on the observed and simulated fields, a threshold of 5 dBZ is applied to both fields in order to avoid that large innovations associated to non-precipitating signals would lead to large analysis increments without physical relevance.

P7L10 What do the authors mean by "on average along the vertical"?

It meant that the whole volume was taken into account in its entirety, calculating errors spanning all the vertical extension of data. The sentence is modified as follows:

Employing all radar data available during the test case, a reflectivity observational error equal to 5 dBZ is estimated, as found also by Tong and Xue (2005).

P7L13 In this section a description of the experimental setting is presented. The clarity of the first two paragraphs needs to be improved. For example some operational systems are mentioned that are not used in the rest of the paper. It would be good to have a comparison between the operational systems and the

experimental system, but in this section only the information regarding the experiments should be included. Table I caption, replace trial by experiment

The operational system is mentioned because several aspects of it (like boundary conditions) are replicated in the experimental set-up. Furthermore, the whole operational set-up is substantially used in the conv60 experiment, the "control" experiment against which we evaluate if the assimilations of reflectivity volumes is advantageous. Anyway to improve the understanding, some small modifications have been made to section 2.4 and, in particular, P7L29 (which in the new version of the manuscript is P8L21) is modified as follows:

In the control experiment, called conv60, the set-up of the operational chain, described in the previous two paragraphs, is replicated. In particular, this means that in conv60 experiment only conventional data are assimilated using KENDA through cycles of 60 minutes and the LHN is performed during the forecast step of each assimilation cycle.

Which is the output frequency of the model for the data assimilation cycle?

The output frequency is equal to the time step of model, that is equal to 18 seconds.

P8L13 from February 3rd to February 7th

Done.

P8L16 new precipitation systems

Done.

Verification: It would be nice to show some examples of how the analysis look like and how the forecast look like in comparison with the observations. This will help to have a general idea on how well the system is working and how accurate the forecasts are.

Even if we understand the point of the reviewer, we do not think that showing a case in which the assimilation of reflectivity volumes improves the accuracy of forecast can really add some useful information to the reader. In fact, as shown by verification scores, case in which a positive impact can be found alternates to some in which the impact is negative. At the same time, we agree with the reviewer who said that the number of figures was already quite large.

The figure in which the areal averaged precipitation is shown is based on the use of dependent data for the verification of the assimilation system. The authors said that since all the experiment are verified in this way this should not be a problem. However for me this is not convincing, since validating with dependent data might not detect issues like overfitting. An analysis closes to the observation is not necessarily the best analysis.

Verification based on areal averaged precipitation employs independent data, since the only observations considered are rain-gauges which are not assimilated. To improve the understanding, some modifications have been done in section 2.5.

P9L15 Describe the SAL acronym

SAL acronym (which derives from the name of components) is now described in the abstract of the paper. In the verification section was already present.

P9L17 Here the authors said that SRI observations are not independent, however the SAL approach is applied to the precipitation forecasts and not to the analysis. In this sense the observations are independent because these observations has not been assimilated yet.

P9L17 and P9L18 are removed

P9L30 Please provide more detailed explanation on this limitation of SAL. My understanding from this paragraph is that SAL can only work with one precipitation system at a time. But this is difficult to guarantee even if the domain is very small as proposed by the authors.

The problem of using SAL in a large domain is that, if precipitating system are strongly different, results will not be representative for the weakest system. To better explain the problem this sentence is added:

In fact, if the domain contains strongly differing meteorological systems, then results obtained using the SAL technique may not be representative for the weakest one.

P10L16 Only results concerning 1 mm are shown. Are the results sensitive to the threshold used in the SAL method? Can the authors comment on the results obtained with other thresholds as well?

Verification with SAL has been done also using a threshold of 3 mm, but results are not included in the manuscript because they are not significantly different from those obtained with a 1 mm threshold. Anyway, we add a sentence in the paper about this (P12L9 of the new version of the manuscript):

Verification using a 3 mm threshold was also performed but, since results do not differ significantly from those obtained with a 1 mm threshold, they are not shown here.

We report here the plots: on the left the ones for a 1 mm threshold (shown in the paper), on the right those with a 3mm threshold.

When comparing conv60 to rad60, rad60_nolhn and rad60_Bm no significant difference can be noticed and the conclusions written in the manuscript are unaffected: the assimilation of reflectivity volumes (applying or not at the same time LHN) does not improve consistently the accuracy of forecast precipitation; instead, using additive inflation slightly worsen results.

[Figure]

When considering the length of assimilation cycles, small differences can be noticed when using a threshold of 3 mm instead of 1 mm. At forecast time +3h, rad15 is the worst while rad30 and rad60 are similar. At +6h, rad30 is slightly better than rad60. Anyway, as stated in the manuscript, shortening the length of assimilation cycles does not affect significantly the accuracy of forecast precipitation.

[Figure]

Finally, regarding the reflectivity observational error (roe), verification with a 3mm threshold (below) leads to same conclusions obtained when considering a 1 mm threshold (above). When a very small value of roe (0.5dBZ) is employed to assimilate reflectivities throughout cycles of 15 min, accuracy of precipitation forecasts is strongly worsened. On the contrary, using a roe=10dBZ does not have a relevant impact.

[Figure]

P12L8 It is not clear how the kinetic energy spectra can be used to identify the effect of the imbalance. May be the evolution of the spectra with the forecast lead time would be a better tool to detect the presence of small scale noise that arises as a consequence of the assimilation of observations (in a similar way as it is done with pressure tendencies).

What we want to assess here is if shortening cycles from 60 minutes to 15 minutes introduces small scale noise in the analyses. Following Skamarock (2004), the fact that KE spectra are identical for the two experiments (especially at the highest wavenumbers) allows us to state that no imbalances are introduced when employing 15 minutes cycles instead of 60 minutes cycles. To improve the understanding of this point, the sentence P12L11 in the original version of the manuscript (P14L7 in the new version of the manuscript) is modified as follows:

Kinetic energy spectra of rad15 (red) and rad60 (green) are almost overlapping, even at very small wavelength, indicating that shortening the length of cycles from 60 to 15 minutes does not introduce imbalances in the analyses(Skamarock, 2004).

**Referee #2**

**General comments:**

I do recognize the authors' efforts on assimilating the radar data in their regional data assimilation system, KENDA. However, I felt that the setups of the experiments cannot clearly illustrate the impact of radar reflectivity on precipitation prediction, given that radar information has been injected into the model state through latent heat nudging. Also, with a high-resolution setup, it is somewhat surprising that there is no benefit from more rapid updates. I am also concerned a potential systematic underestimation of precipitation (and humidity) when a strong dependence on radar data is tested. These seem to lead to issues of radar data quality or how the authors handle the raw radar data. Based on these concerns, I will recommend major revision for this manuscript and hope the authors can address the following comments in their revised manuscript.

**Major comments:**

1. I understand that the assimilation configuration used in this study attempts to be close to the operation settings as much as they could. However, a big question is whether the justification of the impact from radar data on precipitation is fair, given that the precipitation nudging is always applied and the observations for verification contains both information of radar and surface rain gauges. Is it possible to conduct experiments without LHN for clean comparison? E.g. an experiment assimilates conventional data only and compares with the experiment that assimilates conventional and radar data. And, compare the results with the rain gauges data?

- Does the result imply that LHN is more effective than radar data assimilation?

As suggested by both the reviewers, an experiment (rad60_nolhn) in which conventional data and reflectivity volumes are assimilated without LHN is added to the manuscript. Details of the set-up employed are described in section 2.4 (as for all the other experiments) while results and the comparison with rad60 (in which LHN is applied together to the assimilation of reflectivity volumes) are provided in section 3.1.

No significant changes can be noticed in the results when comparing rad60 and rad60_nolhn. This means that LHN is no more effective than radar data assimilation. Therefore, it is decided to not switch off the LHN for the other experiments. In fact, this choice does not affect the results of the sensitivity tests that are presented in this work and, at the same time, the LHN allows to use radar derived information on the state of the atmosphere in the whole Italian country, despite reflectivity volumes can be assimilated, at present, only over Northern Italy.

Some modifications and additions are made in the abstract, in the introduction and in the conclusions to better empathize the use of LHN in combination to the assimilation of reflectivity volumes. Furthermore, it is explicitied that a test is performed to evaluate the impact of switching off LHN when reflectivity volumes are assimilated.

Regarding verification, we stress that areal average precipitation (Figure 4 in the new version of the manuscript) is computed during the assimilation cycles and using rain-gauges as observations (independent data, since they are not assimilated). Regarding SAL verification, it is applied to the forecasts. In this sense the observations (radar estimation corrected by rain-gauges) are independent because these observations has not been assimilated yet.

2. Intuitively, the assimilation of radar data is expected to improve the precipitation. It doesn't seem to be a reasonable choice to me that only examining the absolute value of the components of SAL, without trying to distinguish the possibility of overforecasting or underestimation of the precipitation.

Although in the manuscript only the absolute value of the components of SAL is shown, we have looked carefully at SAL values for each experiment and forecast. Only when significant results are found (like the underestimation of average precipitation in rad15_roe0.5 experiment) they are reported in the manuscript. Therefore we don't think that there is a way to show results (synthetically) better than the one we provided.

- In most of the literate using radar data, the impact is mostly seen in the first 6-h forecast and some even only last for 3 hours. Do the authors see a clear impact for the 1-h or 2-h lead time?

  When considering precipitation accumulated hourly and applying a 1 mm threshold for the

verification (see figure below), a clear positive impact when assimilating reflectivity volumes can be noticed at forecast lead time +2h, while a small positive impact can be noticed at +1h and +3h (in this case, especially for rad60 that is when LHN is applied together to the assimilation of reflectivity volumes). In this article, we focused on longer forecast lead times because they are much more interesting from an operational point of view.

[Figure]

3. It is not too surprising to me that rad60_BM has a worse performance since the perturbations used to augment the ensemble-based background error covariance may be in larger scale. I will suggest either remove this experiment or illustrate the reasons that degrades the performance.

Actually, using the same set-up (perturbations from the ICON model into regional data assimilation with COSMO and KENDA) has been gainfully employed at MeteoSwiss (Leuenberger, D. and Merker, C.: "Additive Covariance Inflation in an operational, convective-scale NWP Ensemble Kalman Filter Assimilation System", Poster contribution at the International Symposium on Data Assimilation (ISDA) 2018, https://isda2018.wavestoweather.de/5 program/poster_presentations/p6_1_leuenberger.pdf.).

Anyway, to improve the understanding, some modifications/additions are made from P5L21 in the new version of the manuscript:

Since Q is not known, it is assumed to be proportional (by a factor smaller than 1) to a static background error covariance B (Mitchell and Houtekamer, 2000). This technique has already been employed with a positive impact in convective scale data assimilation (e.g. Dowell and Wicker, 2009). In the present work, additive inflation is used together with multiplicative inflation and to RTPP only in one experiment, employing a climatological B-matrix from the 3D-VAR of the Icosahedral Nonhydrostatic (ICON) global model (Zängl et al., 2015). Although the use of a lower resolution B-matrix may not allow to properly characterize the model error at the smallest scales, the same configuration has been gainfully employed at MeteoSwiss (Leuenberger and Merker, 2018).

4. It is unclear to attribute the degradation of using a sub-hourly assimilation window to location of rainfall nuclei (Page 13, line, 4). Can the authors explain why a more rapid update (15 or 30-min window) lead to a worse performance than the one using a 60-min window since using a short assimilation window does not introduce the imbalance issue?

When we noticed the small degradation of performance associated to shorter cycle, we thought that the cause might be the application of some balance constraints (hydrostatic and saturation adjustments) when computing analysis. With shorter cycles they are applied more frequently. So, we decided to remove them

but results were almost unaffected. Therefore, it is not yet clear to us why results degradates but we want to stress that the degradation is really small and when considering different threshold (like 3mm) for the verification the degradation becomes even smaller.

5. The authors explain that a larger A component in SAL with the use of small observation error (roe0.5) is due to a systematic underestimation of the average precipitation over the domain or as the example showing a result of decreased humidity. With a strong confidence in observations, such results will be highly dominated by the characteristics of the radar reflectivity data. Do the authors observe that the rain estimated by radar data is underestimated as compared with the rain gauge data? Is there a calibration issue such as the attenuating effect in radar data or the QC procedure (O-B > 5dBz) before the superobservations were constructed?

Raw radar volumes are pre-processed before their use in the KENDA system. Volumes come from different Regional Meteorological Services and they are subject to different cleaning/calibration procedures. Corrections take into account clutter, beam blocking and attenuation. Unfortunately, as explained to the other reviewer, these procedures are  described, if present, by internal documentation in italian, not suitable for being used as a reference.

The "procedure" O-B>5dBZ is not a quality check, it is explained more in detail in the next replies section regarding set-ups about radar data assimilation.

However the text was not clear and the sentence P7L4 in the new version of the manuscript has been modified:

Before assimilation raw reflectivity are pre-processed taking into account non meteorological echoes, beam blocking and attenuation to improve the quality of data.  In particular, it is important to eliminate the clutter signal…

- In the experiments of rad60_roe10 and rad60_roe0.5, is the QC during assimilation the same?
  Yes, the QC is the same.
- I don't quite follow with the rationale in the last paragraph on Page 15. With rad15_roe0.5, It should be the assimilation introduces the small-scale features, instead of losing the ability to "correct" the small-scale errors. To verify this, I suggest that the authors can compare the KE spectrum before (background) and after (analysis) assimilation.
  In the paragraph the expression "small scale structures" was used wrongly to indicate "small scale noise". Now it is corrected. We stress that, analyzing KE spectra, we only want to assess if some configurations employed to assimilate reflectivity volumes provide more balanced analyses than other configurations. According to Skamarock (2004), higher values of KE at the smallest wavelengths of experiment rad15_roe0.5 compared to those of rad60 indicate that the former experiment is more unbalanced than the latter.

6. Information and setups about Radar data assimilation are not clear.
- Although Bick et al. (2016) had described the operator characteristics, and other radar data management. It will still be essential for this paper to briefly provide the important information related to data assimilation. For example, the volume used to construct the superobservation (degree, gate, etc..?). Are all the radar data from different observation intervals with different radars used for constructing the superobservations?
  Informations regarding volume data in input  are given in the description in the section "assimilated data": volumes have a range resolution of 1km, while the azimuthal resolution is 1 degree for Bric della Croce and Settepani and 0.9 degree for San Pietro Capofiume and Gattatico. Superobservations are made individually for each acquisition.
- Page 7, line 6: Is there a precondition to reject (O-B > 5dBz) to avoid large innovations associated to non-precipitating signals? If (O-B >5dBz), doesn't it mean that observation tend to have more rain water? Are the assimilation/forecast results sensitive to such choice?
  To avoid misunderstandings, we stress that what the reviewer calls "O-B > 5dBz" means that reflectivity values which are smaller than 5 dBZ are set equal to 5 dBZ. This correction is made for

both the observation and background fields. This is done to avoid that large innovations associated to non-precipitating signals would lead to large analysis increments without physical relevance. This choice does not imply that observation field tend to have more water, since the "correction" is applied to both observation and background fields. We have not tested different values of threshold, since the value of 5 dBZ has already been employed in other studies (Bick et al., 2016).

- If the horizontal grid-spacing of the analysis domain is 2.2km, isn't it too coarse to have superobservations with resolution of 10km?
  Actually, analysis weights are computed on a coarser grid (6.6km). Anyway, even if the analysis grid had been 2.2 km, a higher resolution for superobbing would have violate the assumption that observations are independent.
  Some explanations about the analysis coarse grid are added at the end of section 2.2:
  The KENDA suite also allows to compute the analysis weights on a coarsened grid (Yang et al., 2009). Weights computed on this coarsened grid are then interpolated to the model grid and afterwards used to calculate analysis increments. In this way, the computational cost is decreased without affecting negatively the accuracy of analysis (Yang et al., 2009). In the present study, a coarsening factor equal to 3 is employed

- Since Bick et al. (2016) used an observation error of 10dBz, is there a particular reason why this study reduces the observation error to 5dBz?
  As stated in the last paragraph of section 2.3, the value of 5 dBZ has been estimated by applying Desroziers' statistics to our case study. To stress this concept, it is stated also at the beginning of section 3.3, modifying the sentence as follows:
  In addition to the value of 5 dBZ employed so far, which was estimated applying the diagnostic described in Desroziers et al. (2005) to this case study, two other values of roe are tested: 10 dBZ and 0.5 dBZ.

- Page 7, line 28: Isn't the radial velocity also assimilated? It's not clear to me why the authors only emphasize on the contribution from reflectivity.
  In this work only reflectivity has been assimilated. The radar operator gives the possibility to assimilate also radial winds, but at the moment we haven't tested them yet. As it does not seem too clear, we have modified the paragraph on P6L11:
  Although the operator gives the possibility to assimilate both radial winds and reflectivities, in the present work only reflectivity volumes are assimilated.

**Minor comments**

Please provide the following Information

- Page 3, line 28: what is the model top of the model?
  This sentence is added:
  The model top is at 22 km.
- Page 5, line 4: please spell out the full name of the ICON model.
  Done.
- Page 5, line11, 14: It's not clear the composite map is composed by what data? Radar only? Or weighted average with the surface rain rate? Is this the same as the observations employed to perform SAL? (Page 9, lines 16-17)
  SRI from the radar composite is composed only by radar data. To perform SAL, SRI (used for LHN during assimilation) is corrected using rain-gauges. Please note that SAL is employed only for the verification of forecasts. Some small modifications are made in the paragraph to improve its clarity.
- Page, 7, line19: I would suggest to cite the original reference for the LETKF scheme: Hunt et al. 2007.
  In Bonavita et al. (2010) the implementation of the LETKF in COMet is described in detail, so we think it is a better reference than Hunt et al. (2007).
- Should I assume that the horizontal grid-spacing of the domain for assimilation is 2.2km?
  No, as reported above, a coarse analysis grid is employed. Details about that are added at the end of section 2.2.

- Page 10, line 20: "observed rainfall field consists of at least 1000 grid points"=> It would be better to change the observed target based on the definition of area (e.g. 50km x 50km?).
  The sentence is modified as follows:
  The average is computed considering only cases in which the observed rainfall field consists of at least 1000 grid points (3 events at lead time +6h, 4 otherwise), which is approximately equal to an area of 50x50 km^2.
- Page 16, line 3-4: Actually, a lot of efforts have been devoted to assimilation of radar reflectivity data already. I am not sure why the authors have such statement
  We refer to the assimilation of reflectivity volumes in an operational data assimilation system. The sentence is modified as follows:

[revised manuscript text omitted]

---

## Author Response (AR2)

**Editor**

**Dear Authors,**

both Reviewers have sent a further report about your second version. You will see that whilst both Reviewers appreciate the relevance of your work and have found satisfactorily your revision, they still have major concerns. In particular they both agree, sometimes in a complementary way, that the paper has to provide further comparisons/results in order to fulfil its goal of analysing the impact of radar data over, for instance, experiments where reflectivity data are not assimilated at all, or on short forecast lead-times.

I also agree with them that, addressing those concerns is necessary and overall will further strength and improve the manuscript.

I hope that you will consider submitting a revised version where the issues arisen by the Reviewers are considered and properly responded. I strongly encourage to do that and I will be happy to receive your revised version and responses.

Dear Editor,

As reported in the answers to the Reviewers, following their indications and reconsidering some of the previous suggestions, we decided to radically revise our manuscript. First of all, the number of forecasts for each experiment has been increased from 5 to 22: forecasts are now initialized each 3 hours from February 3 at 12 UTC to February 6 at 06 UTC instead of each 12 hours from February 4 at 00 UTC to February 6 at 00 UTC. In this way, results are much more significant from a statistical point of view. Furthermore, verification is performed considering hourly (instead of 3 hourly) precipitation to evaluate if the assimilation of radar volumes has an impact in the first few hours of forecast. Moreover, verification has been extended by introducing the Fractions Skill Score (FSS) to compare the QPF of the experiments. This score is applied to the precipitation field over the whole Italian country employing a 1 mm and a 5 mm thresholds. Finally, an experiment in which only conventional data are assimilated without LHN (conv60\_nolhn) has been added to evaluate the case in which no radar information is employed in the data assimilation system. At the same time, the experiment rad60\_Bm has been removed since we think that further investigation is needed to properly evaluate the impact of the additive inflation.

Due to these modifications, some results are changed. The assimilation of reflectivity volumes improves QPF accuracy compared to the conventional set-up (conv60) and to conv60\_nolhn, even if the impact lasts only for the first few hours of forecast. Furthermore, the assimilation of observations collected in the last 15 minutes of each assimilation cycle (rad60\_lst15) further enhances slightly the improvement. Other results are similar to those observed in the previous version of the manuscript but now they are much more significant from a statistical point of view: the assimilation of reflectivities with or without LHN does not substantially affects results, the shortening of the cycle length reduces QPF accuracy and the use of a too small value of the reflectivity observational error introduces imbalances which dramatically worsens results.

Finally, as a consequence of the modifications, some changes have been made in the introduction and in Section 2. Moreover, verification has been reorganized in a new section (section 3) while result and conclusions sections have been radically modified.

1) the authors might want to relate the general context of their study to a recent essay work: Yano et al., 2018: Scientific challenges of convective-scale numerical weather prediction. BAMS <a href="https://doi.org/10.1175/BAMS-D-17-0125.1">https://doi.org/10.1175/BAMS-D-17-0125.1</a>.

Done.

2) I suggest the authors to include a brief mention of the conclusion they are drawing at the end of the abstract. In its present form it well describes the problem tackled and the methodology therein, but not much on the authors' main conclusion.

Done. These sentences are added at the end of the abstract:

Results show that the assimilation of reflectivity volumes has a positive impact on QPF accuracy in the first few hours of forecast both when it is combined to LHN or not. The improvement is further slightly enhanced when only observations collected close to the analysis time are assimilated, while the shortening of cycles length worsens QPF accuracy. Finally, the employment of a too small value of roe introduces imbalances in the analyses resulting in a severe degradation of forecast accuracy, especially when very short assimilation cycles are used.

3) in relation with the somehow unexpected lack of improvement when the data assimilation interval is further reduced, the authors might want to relate/interpretate it based on the study on dynamical instabilities within a convective-resolving model by Uboldi and Trevisan, 2015: Multiple-scale error growth in a convection-resolving model. Nonlinear Processes in Geophysics 22, 1–13, 2015.

This sentence has been added to the conclusion:

A possible explanation, which needs further investigation, is that the more frequent assimilation reduces the analysis error making the ensemble too small to properly characterize the forecast error (Uboldi and Trevisan, 2015).

**Referee #1**

The authors did a great effort to address my comments. I found that the paper has improved, however some important aspects are still unclear.

**Major points**

One of the main objectives of this paper is to assess the impact of assimilation of reflectivity data in a convective scale data assimilation system. However all the experiments presented in this paper assimilate reflectivity data either using nudging or using direct reflectivity assimilation. An experiment in which only conventional observations are assimilated would be very useful to properly assess the impact of assimilating reflectivity. If this experiment is performed then it would be possible to evaluate if nudging and direct reflectivity assimilation provides equivalent results or if we are in a case in which the assimilation of reflectivity do not produce a significant impact upon the forecast skill. In the later case it would also interesting to provide some hypothesis about the causes for such behavior (for example if the complex terrain over the study region is significantly enhancing rain predictability reducing the impact of initial conditions at the mesoscale over the short range forecast skill, or perhaps the impact of reflectivity upon forecast skill is limited to lead times under 3 hours and is not clearly seen beyond that time). The sensitivity experiments will also became more meaningful if the authors can demonstrate that including reflectivity data is producing a positive impact upon forecast skill in this case.

Following the indications of both referees and reconsidering some of the previous suggestions, we decided to radically revise our manuscript. First of all, the number of forecasts for each experiment has been increased from 5 to 22: forecasts are now initialized each 3 hours from February 3 at 12 UTC to February 6 at 06 UTC instead of each 12 hours from February 4 at 00 UTC to February 6 at 00 UTC. In this way, results are much more significant from a statistical point of view. Furthermore, verification is performed considering hourly (instead of 3 hourly) precipitation to evaluate if the assimilation of radar volumes has an impact in the first few hours of forecast. Moreover, verification has been extended by introducing the Fractions Skill Score (FSS) to compare the QPF of the experiments. This score is applied to the precipitation field over the whole Italian country employing a 1 mm and a 5 mm thresholds. Finally, an experiment in which only conventional data are assimilated without LHN (conv60\_nolhn) has been added to evaluate the case in which no radar information is employed in the data assimilation system. At the same time, the experiment rad60\_Bm has been removed since we think that further investigation is needed to properly evaluate the impact of the additive inflation.

Due to these modifications, some results are changed. The assimilation of reflectivity volumes improves QPF accuracy compared to the conventional set-up (conv60) and to conv60\_nolhn, even if the impact lasts only for the first few hours of forecast. Furthermore, the assimilation of observations collected in the last 15 minutes of each assimilation cycle (rad60\_lst15) further enhances slightly the improvement. Other results are similar to those observed in the previous version of the manuscript but now they are much more significant from a statistical point of view: the assimilation of reflectivities with or without LHN does not substantially affects results, the shortening of the cycle length reduces QPF accuracy and the use of a too small value of the reflectivity observational error introduces imbalances which dramatically worsens results.

Finally, as a consequence of the modifications, some changes have been made in the introduction and in Section 2. Moreover, verification has been reorganized in a new section (section 3) while result and conclusions sections have been radically modified.

**Minor points**

Page 3, line 25 ((LHN - Remove the extra parenthesis. Done.

Page 7, Why 3D data is available only for radars over Northern Italy? Is because how data is archived?

Meteorological radar in Italy belong to different institutions. For the case study presented in this manuscript, only those of our institution (Arpae Emilia-Romagna) and of Arpa Piemonte were available to us. Anyway, note that in the period examined (3-7 February 2017) precipitations mainly interested Northern Italy.

Page 8, line 10 observed precipitation Done.

Page 8, line 23 every 3 hours Done.

Page 8, line 26 and Done.

Page 10, line 7 February 5 line 8 February 6. Done.

Page 10, Line 7 On February 5 / new precipitating systems Done.

Page 10, Line 15 Spatially averaged forecasted precipitation is compared against ... Done.

Page 10, Line 29 precipitating systems instead of precipitation nuclei Done.

Page 12, Line 18 reflectivity data is The sentence, as well as the entire section, has been completely modified.

Page 12, Line 20 Spatially averaged The sentence, as well as the entire section, has been completely modified.

Page 18, lines 20-21 The experiment assimilating only conventional data is not described in this work. All the experiments assimilate reflectivity either using nudging or direct assimilation. This is explained in the followings entences, but it would be better to state the conclusion in a clearer way from the beginning instead of makingthis clarification later.

The sentences, as well as the entire section, have been completely modified. Note also that now an experiment in which only conventional data are assimilated (conv60\_nolhn) has been discussed in the manuscript.

Page 19, For the case study considered in this work... Since the experiments provided in this work shows little sensitivity to the different strategies used to assimilate reflectivity, the lack of sensitivity reported here might be a consequence of the general lack of sensitivity of these forecast to the assimilation of reflectivity.

As stated previously, to investigate this lack of sensitivity, in the new version of the manuscript the verification of results has been radically enhanced by considering much more forecasts, by implementing FSS and by applying both SAL and FSS to hourly precipitation. It is still true that the impact of the assimilation of reflectivity volumes is not extremely marked, but however it is present in the first few hours of forecast.

To further reduce the number of figures, Figure 1 and 2 can be combined if the color scale is changed accordingly.

Done.

I still think that including some examples of the precipitation or reflectivity forecast in the paper would be useful. Statistical metrics are powerful and useful, but having such examples also gives a good subjective idea of how good is the fit between the forecast and the observations. This is particularly useful in the context of novel high resolution convective scale data assimilation systems. If the results changes from case to case, then two examples can be provided.

Hourly maps of precipitation for the first 3 hours of forecasts initialized on February 3 at 12 UTC are shown in Figure 4.

**Referee #2**

Based on the evidence presented in this study, it is difficult to draw a solid conclusion about positive impact from assimilating radar reflectivity. I think it is mainly because the authors are focusing on the one-day QPF, which is claimed to be the operational interest. However, it is well recognized by the literature that the impact of radar data is identified for (very) short-term forecast. I would like to suggest the authors that since this work is the new component for the KENDA system, it is essential to identify and justify the impact of the radar data, even though the impact may only last few hours. Also, I'd think that it is important to explain why additionally assimilating radar data doesn't seem to gain more benefit than applying LHN. Is this because the information is double used or redundant? Based on these concerns, I suggest major revision for this manuscript.

Following the indications of both referees and reconsidering some of the previous suggestions, we decided to radically revise our manuscript. First of all, the number of forecasts for each experiment has been increased from 5 to 22: forecasts are now initialized each 3 hours from February 3 at 12 UTC to February 6 at 06 UTC instead of each 12 hours from February 4 at 00 UTC to February 6 at 00 UTC. In this way, results are much more significant from a statistical point of view. Furthermore, verification is performed considering hourly (instead of 3 hourly) precipitation to evaluate if the assimilation of radar volumes has an impact in the first few hours of forecast. Moreover, verification has been extended by introducing the Fractions Skill Score (FSS) to compare the QPF of the experiments. This score is applied to the precipitation field over the whole Italian country employing a 1 mm and a 5 mm thresholds. Finally, an experiment in which only conventional data are assimilated without LHN (conv60\_nolhn) has been added to evaluate the case in which no radar information is employed in the data assimilation system. At the same time, the experiment rad60\_Bm has been removed since we think that further investigation is needed to properly evaluate the impact of the additive inflation.

Due to these modifications, some results are changed. The assimilation of reflectivity volumes improves QPF accuracy compared to the conventional set-up (conv60) and to conv60\_nolhn, even if the impact lasts only for the first few hours of forecast. Furthermore, the assimilation of observations collected in the last 15 minutes of each assimilation cycle (rad60\_lst15) further enhances slightly the improvement. Other results are similar to those observed in the previous version of the manuscript but now they are much more significant from a statistical point of view: the assimilation of reflectivities with or without LHN does not substantially affects results, the shortening of the cycle length reduces QPF accuracy and the use of a too small value of the reflectivity observational error introduces imbalances which dramatically worsens results.

Finally, as a consequence of the modifications, some changes have been made in the introduction and in Section 2. Moreover, verification has been reorganized in a new section (section 3) while result and conclusions sections have been radically modified.

**My commentsare as follows:**

• Based on the verification merits used in this study, it is difficult to justify the impact of radar data. Have the authors consider adopting other verification merits such as POD, FAR, TS and BIAS to identify whether there is clearer signal about the data impact? Or, illustrate the results with a case study (from the 4-day assimilation)?

As indicated previously, we decided to implement FSS, a completely different score compared to SAL (it is also applied over a larger domain) which, at the same time, allows to overcome the problems of traditional grid-point based score like POD, FAR, Ts etc. The FSS score is explained in Section 3 and results are shown in Section 4.

• Discussion about Fig. 4 is vague. Line13, Page11: "Overall, the correspondence of rad60 .... to observations is equal or better than that of conv60 (red)". Also, by eye, rad60\_nolhn is slightly better than rad60. rad60\_Bm tends to have the least rainfall among all the rad60-related experiments shown on Fig. 4 but the authors claim that the impact of the additive inflation cannot be judged. To quantify such statement. I suggest to summarize Fig. 4 in terms of RMS error.

The RMSE error has been introduced to evaluate the correspondence between experiments and observations. Note also that the experiment rad60\_Bm has been removed and the section radically modified.

• As mentioned in my general comments, I'd think that it is important to explain why additionally assimilating radar data doesn't seem to gain benefit than applying LHN. Is this somewhat related to the fact that the SRI product uses the radar reflectivity? The authors can illustrate the result with even a case demonstration. Should we expect a large difference between radar data assimilation and LHN with heavy rainfall events? As suggested by the referee, verification is now applied to hourly precipitation and FSS score has

been added, Thanks to this, is possible to assess that the assimilation of reflectivity volumes improves QPF accuracy between lead time +2h and +4h compared to the assimilation of only conventional observations combined to LHN (conv60).

- Although the authors focus on the impact of radar data on 1-day QPF, it is undeniable that the impact is mostly evident within 6 hours. Therefore, I suggest that the authors should address the impact of radar data on short-range forecast (< 6h). Done, considering a forecast range of 8 hours.
- It is concluded that the radar data only slightly improve QPF both during the assimilation procedure and for the subsequent forecasts, compared to the assimilation of only conventional data (Line3, Page 18). This sentence should be modified for 1-day QPF. Done. Note that the whole section has been radically modified.
- Why does rapid update (15-min) with a small observation actually lead to a dry condition over northern Italy (Fig. 9)? Fig. 9 should be also compared with results from rad15 with rad15\_roe0.5.

We decided to substitute rad60 with rad15 in the figure. As stated previously, the differences between rad15 and rad60 are very small but we agree with the referee that it make more sense to compare rad15\_roe0.5 to rad15 rather than to rad60. We decided to not put both rad15 and rad60 in the same figure because, otherwise, it would have been unreadable.

The dry conditions over Northern Italy are, in our opinion, an effect of the big amount of imbalances in the analyses, as verified by evaluating KE spectra. However, how this results in a reduction of humidity is still an open problem. Anyway, please keep in mind that rad60\_roe0.5 is an unrealistic experiment (the value of roe is extremely small), performed only to evaluate the sensitivity of results to the value of roe.

**Minor comments:**

Since radar reflectivity is the main focus in this work, the authors should briefly comment or summarize whatare the issues/difficulties with assimilation of reflectivity volumes in the introduction.

Problems associate to the assimilation of radar data are explained, approximate lively, between P2 L35 and P3 L23

"Grater" is often used in the text. But, I think the authors meant "greater". Done.

[revised manuscript text omitted]
| rad60 –nolhn          | 60                  | conv. + radar              | 5         | $\frac{No LHN}{\sim}$                      |
| rad60_ <mark>Bm-nolhn</mark> | 60                  | conv. + radar              | 5         | Additive inflation No LHN                  |
| rad30                        | 30                  | conv. + radar              | 5         | -                                          |
| rad15                        | 15                  | conv. + radar              | 5         | -                                          |
| rad60_lst15                  | 60                  | conv. + radar              | 5         | Use obs. in the last 15 min. of the window |
| rad60_roe10                  | 60                  | conv. + radar              | 10        | -                                          |
| rad60_roe0.5                 | 60                  | conv. + radar              | 0.5       | -                                          |
| rad15_roe10                  | 15                  | conv. + radar              | 10        | -                                          |
| rad15 roe0.5                 | 15                  | conv. + radar              | 0.5       | _                                          |

**Table 1.** Experimental set-up of each experiment including the length of the assimilation cycles, the type of observations assimilated, the reflectivity observation error (*roe*) associated to radar data and any additional feature.

- 5 Since the estimation of observation error is not straightforward and different techniques can be applied, it is worth to evaluate the sensitivity of the assimilation system to this parameter. In addition to the value of 5 dBZ employed in the previous experiments, two other values are selected: 10 dBZ or 0.5 dBZ. Both of them are tested employing a 60 minutes assimilation window (*rad60\_roe10* and *rad60\_roe0.5*) and using 15 minutes cycles (*rad15\_roe10* and *rad15\_roe0.5*).
- The experiments described above are carried out over a period of almost 4 days from February 3rd at 06 UTC to February 7th at 00 UTC in 2017. During 3 and 4 February, middle tropospheric circulation over Northern and Central Italy was dominated by southwesterly divergent flows associated with the passage of some precipitating systems. In 5 February On February 5 a trough moved from France to Italy and this caused the formation of new precipitations precipitating systems in Northern Italy. During 6 February February 6 the trough moved slowly from Central Italy to the southern part of the country and precipitation systems weaken gradually. For each experiment, a set of 5 analyses of the deterministic member are used to initialize forecasts
- 15 up to 48 hours is initialized using the analyses generated during the assimilation procedure. Initialization times employed are 04 February at 00 and 24 hours each 3 hours from February 3 at 12 UTC , 5 February at 00 and 12 UTC and to February 6 February at 00. 
[revised manuscript text omitted]